# Calculation of external climate costs for food highlights inadequate pricing of animal products

Maximilian Pieper [1]✉, Amelie Michalke[2] & Tobias Gaugler [3]

Although the agricultural sector is globally a main emitter of greenhouse gases, thorough economic analysis of environmental and social externalities has not yet been conducted. Available research assessing agricultural external costs lacks a differentiation between farming systems and food categories. A method addressing this scientific gap is established in this paper and applied in the context of Germany. Using life-cycle assessment and meta-analytical approaches, we calculate the external climate costs of foodstuff. Results show that external greenhouse gas costs are highest for conventional and organic animal-based products (2.41€/kg product; 146% and 71% surcharge on producer price level), followed by conventional dairy products (0.24€/kg product; 91% surcharge) and lowest for organic plant-based products (0.02€/kg product; 6% surcharge). The large difference of relative external climate costs between food categories as well as the absolute external climate costs of the agricultural sector imply the urgency for policy measures that close the gap between current market prices and the true costs of food.

[1] Technical University of Munich (TUM), Munich, Germany. [2] University of Greifswald, Greifswald, Germany. [3] University of Augsburg, Augsburg, Germany.
✉email: max.pieper@tum.de

Social and environmental costs from the emission of greenhouse gases (GHGs) are currently not considered in the cost structure of farmers or the subsequent food chain[1,2], and are thus a burden on other market participants, future generations, and the natural environment. These external costs are not yet included in the market prices for food and, in the absence of current compensation payments, lead to significant market price distortions[3] and welfare losses for society as a whole[4,5]. In order to close the gap between the current market prices and the true costs of foodstuff, GHG emissions from agriculture have to be quantified and monetized. The United Nation's (UN) polluter-pays principle[6] implies that in order to compensate for externalities, external costs should be levied on the producer prices of food, or other economic policy measures should be taken to reduce or compensate harmful costs caused by food production[7].

There has been some scientific engagement previously, as Pretty et al.[8] set the scene for agricultural externality analysis at this century's beginning: they were able to record significant environmental impacts of agriculture at the overall societal level in monetary terms for the UK. This approach was translated for other regions subsequently, with calculations of agricultural external costs for the USA and Germany[2,9]. However, these first external cost assessments, with their characteristic top-down approaches, did not link specific causal emission values with said costs. Yet, a bottom-up approach for monetizing externalities of country-specific agricultural reactive nitrogen emissions was later developed[10] and subsequently used for an external cost assessment of Dutch pig production[11]. Despite, assessments concerning important agricultural emissions comprehensively differentiating between a variety of food categories are yet missing. There exists a range of studies that quantify food-category-specific GHG emissions[12–15] while other studies disclose the difference of climate effects from conventional and organic practices[16–28]. Monetizing such emissions, however, has been done for constituent food categories only[29]. An encompassing connection between the quantification and monetization of GHG emissions differentiated by food categories and farming systems is what seems to be lacking in the currently available literature.

Congruent to methodological differences for monetizing agricultural greenhouse gases, there are also differences in the estimation level of greenhouse gas costs. Prices per tonne of emission at the stock market, for example, are as low as 5.34 € on average during this study's reference year, whereas they were more than 10 € higher on average ten years prior and have risen up to about 25 € on average especially in the past two years[30]. The German Federal Environmental Agency's (UBA) suggestion for the damage costs of GHG emissions also rose within the last years: in 2010 they suggested a rate of 80 € per tonne of $CO_2$ equivalents (eq)[31], whereas this increased to 180 € per tonne in 2019[32]. This price factor is congruent with the IPCCs evaluation from 2014, which states a reasonable cost rate of 181 $ per tonne of $CO_2$ equivalents, calculating to ~173.5 €/t$CO_2$ eq[33]. This implies that a scientific consensus has been reached over the past years, considering an adequate cost rate for GHG-related damage. Furthermore, the price is expected to rise in the future, whereby a cost rate of over $400 per tonne might be necessary by mid-century[34].

The aim of this paper, by building on previous work, including our own earlier research efforts[35,36], is to provide a method for a differentiated quantification and monetization of GHG emissions of a variety of foodstuff and farming practices. We thereby illustrate the present price difference between current producer prices and true costs. The established framework is tested in the German context and is further applicable for other country contexts and different externalities: Life-cycle assessment (LCA)

tools, such as the one used in this study (see the section "Input data for quantification") for quantifying emissions of the examined foodstuff, also offer the data for other externalities. Further, production quantities as well as producer prices are largely available for other regional contexts. Thereby applicability and transferability of the presented method of quantification and monetization are ensured.

LCA has developed as a commonly used tool for examining material and substance flows of diverse products. Its origins lie in the analysis of energy flows, but it is now commonly used to assess various processes[37]. In general, the LCA method examines environmental and social impacts that occur during the entire lifetime of a product and can involve a monetization of such impacts. This includes both impacts from production and impacts occurring during the usage phase of a product up to its disposal (or consumption), as well as all intermediate emissions[38].

Additional to the consideration of $CO_2$ emissions, all so-called $CO_2$ equivalents (methane, $CH_4$; nitrous oxide, $N_2O$) are considered in greenhouse gas-emission assessments of the current literature, as these gases not based on carbon still contribute to climate effects[39]. These gases each have a defined global warming potential (GWP). Especially during the production of animal-based foodstuff, livestock-related gases, such as methane or nitrous oxide, significantly contribute to the overall GHGs emitted[40].

$CO_2$ is produced in agriculture through microbial degradation (rotting) and the burning of plant waste. In addition, considerable amounts of $CO_2$ previously bound in soils are released into the atmosphere through agricultural processes[41]. Indirect $CO_2$ emissions from agricultural transport, heat generation, and emissions from the production of nitrogen fertilizers[42] are of quantitative relevance as well. $CH_4$ is produced during the composting or conversion of organic substances in oxygen-poor environments, i.e., mainly during the digestion of ruminant farm animals[41]. $N_2O$ is produced in agriculture mainly due to direct emissions from agricultural soils, mostly caused by the over-application of nitrogen fertilizer, and indirect emissions from the production of such fertilizer[43].

Consequently, we develop a calculation of the monetary valuation of carbon footprints for foodstuff, resulting in food (category)-specific external costs. We differentiate between the categories of conventional and organic products as well as animal-, dairy, and plant-based products, but also narrower categories such as beef (animal-based), milk (dairy), or cereal (plant-based). Our analysis shows that external cost differences are especially large between food categories, whereby animal products are associated by far with the highest external costs, followed by dairy and plant-based products. In contrast to food categories, the influence of production methods on external climate costs is much smaller.

If the resulting costs are addressed by economic policies in line with common economic theory, they would enable agricultural externalities to be internalized according to the polluter-pays principle and at the same time strengthen sustainable consuming behavior. Pricing of food that includes environmental and social costs would thus also significantly contribute to fair market conditions, and simultaneously to climate change mitigation.

## Results

**Outline**. The quantification and monetization of externalities from agricultural GHG emissions for Germany is derived in the following. First, the input data are displayed. Second, these data are applied to our methodology (cf. "Method and data" section). Lastly, the output data are derived.

**Quantification**. Using the input data for quantification (for definition and origin refer to the section "Input data for quantification") as starting points, this subsection shows results of the emissions data for food categories at different aggregation levels. All foodstuffs are divided into plant-based, animal-based, or dairy products classified as broad categories. The narrow categories are more fine-grained and divide plant-based foods into vegetables, fruits, cereals, root crops, legumes and oilseed, and animal-based foods into eggs, poultry, ruminants, and pork. Only milk is considered within the dairy products, as processing steps beyond the farmgate would be necessary to achieve other dairy products, such as cheese or butter. This, however, does not fall into the defined system's boundaries, which we chose as cradle to farmgate (cf. "Method and data").

The food-specific conventional emission data $g_{b,n,i,conv}$ is derived from the material-flow analysis tool GEMIS (Global Emission Modell of Integrated Systems)[44] and is the basis for calculating external costs. However, land-use-change-emissions (LUC) are not included in this dataset. Thus, we calculate these emissions ourselves, following the methodology of Ponsioen and Blonk[45] (see the section "Input data for quantification" for a detailed description) for the food-specific, narrow as well as broad categories, but only for conventional production. This is because LUC emissions almost entirely originate from the cultivation of imported crops, from countries where arable land is expanding at the cost of natural land. Only in conventional production, it is unreservedly allowed to import crops (as fodder) from locations outside of the regional context. This is in contrast to organic production where the majority of the fodder must come from farms from the same or directly neighboring federal states[46]. As LUC emissions do currently not arise within Germany (total area of arable land is decreasing)[47], it can be assumed that LUC emissions of organic production (in Germany) are of negligible scope (for details, refer to the section "Method and data").

In order to derive emission data for organic production, the conventional emission data (excluding LUC emissions) is differentiated according to the method described in the "Method and data" subsection on output data resulting in the values shown in the columns for organic production in Table 1.

The results of this differentiation of the GEMIS data are laid out in Table 2, where the emission difference between both systems is calculated for each of the three broad categories (plant-based, animal-based, dairy).

As can be seen in Table 2, the choice of the farming system has the largest effects in the production of animal-based foodstuff. In this category, organic production causes 150% of emissions from conventional production. It is important to note that emissions from LUC are not yet included in the underlying data and calculation, which when considered changes the results for animal-based foodstuff drastically (compare column conv with LUC in Table 1). In the two other broad categories, organic causes fewer emissions than conventional production. Organic plant-based products cause 57% and dairy products 96% of emissions from conventional products. Explanations for these differences are elaborated in the "Discussion".

We aggregate GEMIS emission data ($q_{b,n,i,conv}$) to narrow ($e_{b,n,conv}$) and broad categories ($E_{b,conv}$) by multiplying the respective emission data with the quantitative production shares of food-specific products in narrow categories and the shares of narrow in broad categories (cf. "Input data for quantification"). From these aggregated conventional emission values, we derive emissions for organic production. For narrow as well as broad categories, the respective conventional emission values are multiplied with the applicable emission differences $D_{b,org/conv}$ (see Table 2). The results are illustrated in Table 1.

Examining the broad categories in the left columns of Table 1, it can be seen that animal-based products cause the highest emissions per kilogram of product at 13.38–13.39, followed by dairy at 1.05–1.33 and plant-based products with 0.11–0.20 kgCO$_2$eq/kg product. Within narrow categories, ruminants cause by far the highest emissions with 36.95–37.37 over all products while legumes cause the lowest emissions with only 0.02–0.03 kg CO$_2$eq/kg product. As follows from Table 2, with LUC emissions included, organically produced food causes fewer emissions in the broad plant-based and dairy categories, while causing slightly higher emissions in the animal category. In the narrow categories, organic production performs worse for eggs, poultry, and ruminants. Explanations for emission differences between the different food categories and the production methods will be addressed in the "Discussion".

**Monetization**. When putting the calculated emission values into monetary units with the emission cost rate from the German Federal Environment Agency (UBA) of 180 € per ton of CO$_2$ equivalents[32,33], their absolute external costs can be derived. The results are shown in Table 3 for conventional and organic farming in columns $C_{b,conv}$ and $C_{b,n,conv}$ as well as $C_{b,org}$ and $C_{b,n,org}$, respectively. When

**Table 1 Emission data for food-specific, narrow and broad categories (following the classification from the German Federal Office of statistics[88]).**

Emission data (in kg CO$_2$eq/kg product)

| Broad categories [b] | Prod. method | | | Narrow categories [n] | Prod. method | | | Food-specific [i] | Prod. method | | |
|---|---|---|---|---|---|---|---|---|---|---|---|
| | Conv. [$E_{b,conv}$] | With LUC | Org. [$E_{b,org}$] | | Conv. [$e_{b,n,conv}$] | With LUC | Org. [$e_{b,n,conv}$] | | Conv. [$g_{b,n,i,conv}$] | With LUC | Org. [$g_{b,n,i,org}$] |
| Plant-based | 0.20 | / | 0.11 | Vegetables | 0.04 | / | 0.02 | Field Vegetables | 0.03 | / | 0.02 |
| | | | | | | | | Tomatoes | 0.39 | / | 0.22 |
| | | | | Fruit | 0.25 | / | 0.14 | Fruit | 0.25 | / | 0.14 |
| | | | | Cereal | 0.36 | / | 0.21 | Rye | 0.22 | / | 0.13 |
| | | | | | | | | Wheat | 0.38 | / | 0.21 |
| | | | | | | | | Oat | 0.36 | / | 0.21 |
| | | | | | | | | Barley | 0.33 | / | 0.19 |
| | | | | Root Crops | 0.06 | / | 0.04 | Potatoes | 0.06 | / | 0.04 |
| | | | | Legumes | 0.03 | / | 0.02 | Beans | 0.03 | / | 0.02 |
| | | | | Oilseed | 1.02 | / | 0.58 | Rapeseed | 1.02 | / | 0.58 |
| Animal-based | 8.90 | (13.38) | 13.39 | Eggs | 1.17 | (1.18) | 1.76 | Eggs | 1.17 | (1.18) | 1.76 |
| | | | | Poultry | 13.16 | (15.81) | 19.80 | Broilers | 13.16 | (15.81) | 19.80 |
| | | | | Ruminants | 24.84 | (36.95) | 37.37 | Beef | 24.84 | (36.95) | 37.37 |
| | | | | Pork | 5.54 | (9.56) | 8.34 | Pork | 5.54 | (9.56) | 8.34 |
| Dairy | 1.09 | (1.33) | 1.05 | Milk | 1.09 | (1.33) | 1.05 | Milk | 1.09 | (1.33) | 1.05 |

Food-specific emission data for conventional production was derived from Global Emissions Model for Integrated Systems (GEMIS)[44] and aggregated to narrow and broad categories with German production data[88]; differentiation between conventional and organic production was derived with a meta-analytical approach (for details refer to the "Method and data" section and Supplementary Note 1 and Table 1); land-use change (LUC) data are approximated to be the LUC emissions of soymeal fodder, emissions of it are calculated with the method of Ponsioen and Blonk[45].
Emission data including LUC emissions are shown in brackets. Source data are provided as a source data file.

**Table 2 Determining the emission difference ($D_{org/conv}$) between organic and conventional production in different countries' contexts through the application of meta-analytical methods.**

| Name | Country | Produce | $D_{org/conv}$ | Relevance | | | | |
|---|---|---|---|---|---|---|---|---|
| | | | | PY | CY | SJR | SUM | WEIGHT |
| **Plant-based** | | | | | | | | |
| Aguilera et al. (2015a)[16] | Spain | citrus, fruits | 49% | 10 | 3 | 10 | 23 | 26% |
| Aguilera et al. (2015b)[17] | Spain | cereals, legumes, veg. | 45% | 10 | 3 | 10 | 23 | 26% |
| Cooper et al. (2011)[18] | UK | crop rotation (no differentiated values for specific crops given) | 42% | 8 | 2 | 2 | 12 | 13% |
| Küstermann et al. (2008)[19] | Germany | arable (no specific crop differentiation/rotation described) | 72% | 7 | 3 | 4 | 14 | 16% |
| Reitmayr (1995)[20] | Germany | wheat, potatoe | 63% | 0 | 1 | 1 | 2 | 2% |
| Tuomisto et al. (2012)[21] | EU | arable (no specific crop differentiation/rotation described) | 36% | 9 | 2 | 5 | 16 | 18% |
| | | | 49% | | | | 90 | 100% |
| | | | **57%** | ↓ × 117% | | | | |
| **Animal-based** | | | | | | | | |
| Basset-Mens; Werft (2005)[22] | France | pig | 95% | 5 | 7 | 6 | 18 | 35% |
| Casey; Holden (2006)[23] | Ireland | beef | 82% | 6 | 3 | 10 | 19 | 37% |
| Flessa et al. (2002)[24] | Germany | beef/cattle | 73% | 4 | 5 | 6 | 15 | 29% |
| | | | 84% | | | | 52 | 100% |
| | | | **150%** | ↓ × 179% | | | | |
| **Dairy** | | | | | | | | |
| Bos et al. (2014)[25] | Netherlands | dairy | 61% | 10 | 3 | 4 | 17 | 24% |
| Dalgaard et al. (2006)[26] | Denmark | dairy | 57% | 6 | 2 | 6 | 14 | 20% |
| Haas et al. (2001)[27] | Germany | dairy | 67% | 3 | 8 | 5 | 16 | 23% |
| Thomassen et al. (2008)[28] | Netherlands | dairy | 65% | 7 | 10 | 6 | 23 | 33% |
| | | | 63% | | | | 70 | 100% |
| | | | **96%** | ↓ × 152% | | | | |

Arrows represent the yield/productivity difference for each category; this difference is then multiplied with the emission difference per ha to derive the emission difference per kg (in bold). PY = publishing year, CY = yearly citations, SJR = SciMago journal ranking, SUM = sum of all three factors, WEIGHT = weighted sums of category.
A more detailed explanation of the studies' specifics including the weighting scheme can be found in the Supplementary Note 1 and Table 1. Source data are provided as a source data file.

these external costs are assessed in relation to their corresponding producer price (pp), the resulting percentage surcharge (Δ) reflects the price increase necessary to internalize the GHG-related externalities arising from food production. Relative results for conventional and organic farming are shown in column $\Delta_{b,conv}$ and $\Delta_{b,n,conv}$ as well as $\Delta_{b,org}$ and $\Delta_{b,n,org}$, respectively. Food-specific products (see Table 1) are omitted in this table since their respective monetary costs and percentage price increases follow the same pattern as the narrow category. Please refer to the "Method and data" section for details of the full calculation methodology and data origin.

For the broad category, the results are visualized in Figs. 1 and 2, where Fig. 1 shows the absolute price increases (in Euro), whereas Fig. 2 shows the relative price increases (in percent).

Following, we explain the broad categories' data further. The narrow categories follow the same narrative overall. Looking at Table 4 and Fig. 1, the external costs of organic plant-based products are clearly the lowest (0.02€/kg product). External costs for conventional plant-based products are about twice as high (0.04€/kg product), although still relatively low compared with the other two broad categories. This shows that even the animal-

**Table 3 Producer prices (pp), external costs (C) and percentage price increases (Δ) for narrow and broad food categories when externalities resulting from greenhouse gas emissions are monetized.**

| Broad categories [b] | Prod. method | | | | | | Narrow categories [n] | Prod. method | | | | | | |
| | Conv. | | | | Org. | | | Conv. | | | | Org. | | |
| | $pp_{b,conv}$ (€/kg Prod) | $C_{b,conv}$ (€/kg Prod) with LUC | $\Delta_{b,conv}$ with LUC | | $pp_{b,conv}$ (€/kg Prod) | $C_{b,org}$ (€/kg Prod) | $\Delta_{b,org}$ | | $pp_{b,n,conv}$ (€/kg Prod) | $C_{b,n,conv}$ (€/kg Prod) with LUC | $\Delta_{b,n,conv}$ with LUC | | $pp_{b,n,org}$ (€/kg Prod) | $C_{b,n,org}$ (€/kg Prod) | $\Delta_{b,n,org}$ |
| Plant-based | 0.14 | 0.04 | 25% | | 0.36 | 0.02 | 6% | Vegetables | 0.69 | 0.01 | 1% | | 1.10 | ~0.00 | ~0% |
| | | | | | | | | Fruit | 0.50 | 0.05 | 9% | | 0.57 | 0.03 | 5% |
| | | | | | | | | Cereal | 0.09 | 0.07 | 72% | | 0.31 | 0.04 | 12% |
| | | | | | | | | Root Crops | 0.08 | 0.01 | 14% | | 0.30 | 0.01 | 2% |
| | | | | | | | | Legumes | 0.02 | 0.01 | 33% | | 0.13 | ~0.00 | 3% |
| | | | | | | | | Oilseed | 0.37 | 0.18 | 50% | | 0.42 | 0.10 | 25% |
| Animal-based | 1.66 | 1.60 *(2.41)* | 97% *(146%)* | | 3.41 | 2.41 | 71% | Eggs | 1.21 | 0.21 *(0,21)* | 17% *(18%)* | | 3.42 | 0.32 | 9% |
| | | | | | | | | Poultry | 1.72 | 2.37 *(2,85)* | 138% *(165%)* | | 2.31 | 3.56 | 154% |
| | | | | | | | | Ruminants | 3.38 | 4.47 *(6,65)* | 132% *(197%)* | | 3.90 | 6.73 | 173% |
| | | | | | | | | Pork | 1.35 | 1.00 *(1,72)* | 74% *(128%)* | | 3.61 | 1.50 | 42% |
| Milk | 0.26 | 0.20 *(0.24)* | 75% *(91%)* | | 0.48 | 0.19 | 40% | Milk | 0.26 | 0.20 *(0,24)* | 75% *(91%)* | | 0.48 | 0.19 | 40% |

Producer prices are calculated by dividing the total amount of producer proceeds for each category (in Euro)[99] with its total production quantity[88,89]; external costs are derived by multiplying emission values from Table 1 with the emission cost rate of 180 €/tCO₂eq; percentage price increases are the ratio of external costs to producer prices; in brackets are the values with land-use change (LUC) emission costs included.
In each broad and narrow category, the highest external costs and percentage surcharge are highlighted in red and the lowest in green. Source data are provided as a source data file.

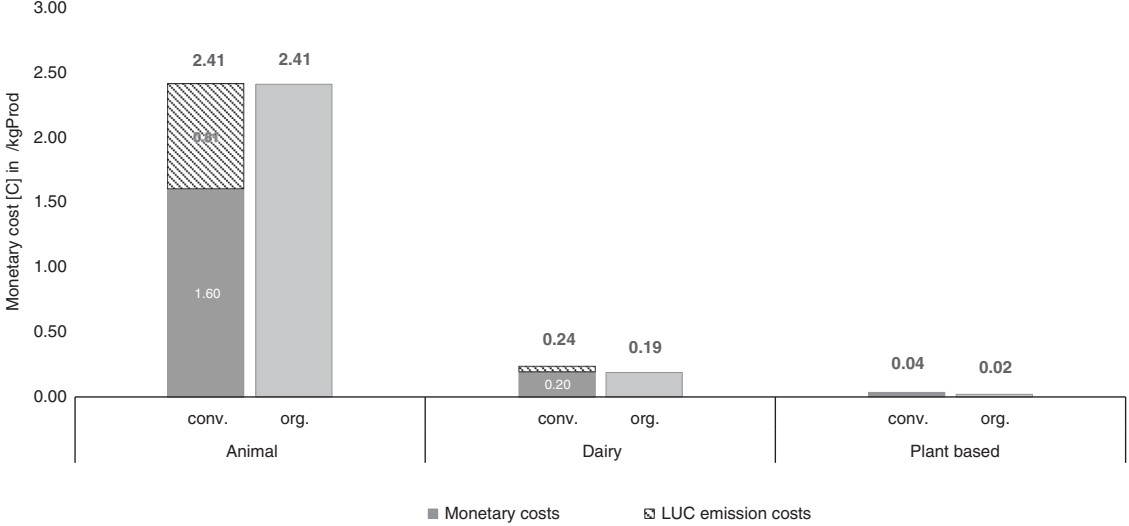

**Fig. 1 Visualization of monetary costs for broad food categories.** Monetary costs [C] for broad categories (animal-based, dairy, plant-based in the comparison between conventional and organic production) arising from monetized externalities of greenhouse gas emissions. For conventional production (animal-based and dairy), the external costs from land-use change (LUC) emissions are highlighted separately. Source data are provided as a source data file.

based product emitting the lowest rate of GHG within its broad category causes higher external costs than the plant-based product emitting the highest rate of GHG emissions within its broad category. Animal-based products cause the highest external costs (2.41 €/kg product), which are 10 times higher than dairy costs and 68.5 times higher than plant-based costs. Here,

conventional farming (2.41 €/kg product) perform as well as organic farming (2.41 €/kg product). In all other broad categories, organic farming outperforms conventional farming. This advantage of organic farming is considerable as it produces 21% less emissions for dairy and 43% less emissions for plant-based products on average per kg.

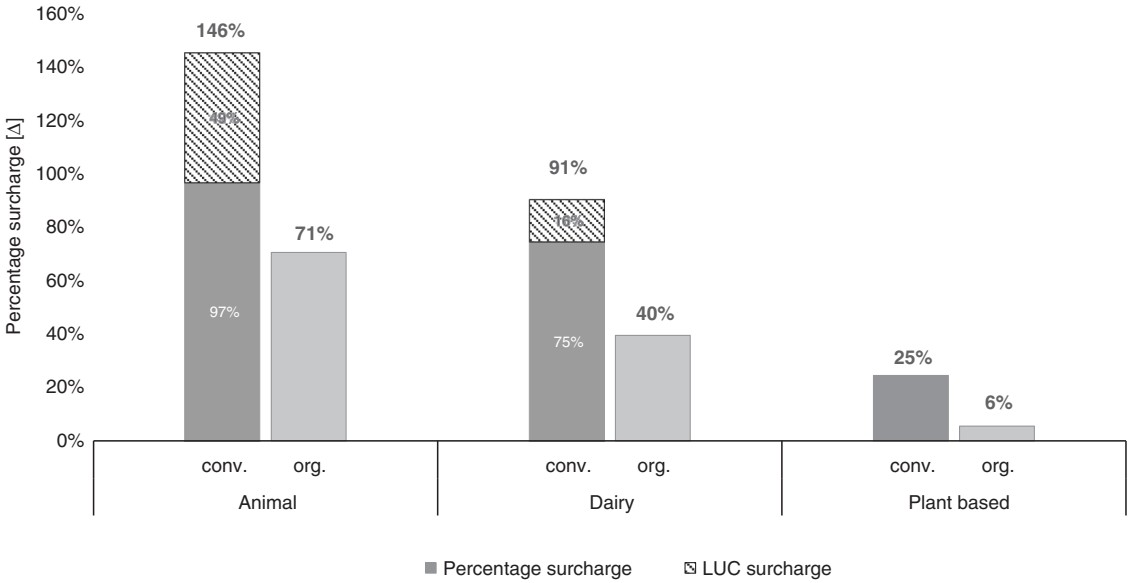

**Fig. 2 Visualization of percentage price increases for broad food categories.** Relative percentage price [Δ] increases for broad categories (animal-based, dairy, plant-based in the comparison between conventional and organic production) when externalities of greenhouse gas emissions are included in the producer's price. For conventional production (animal-based and dairy), the surcharge from land-use change (LUC) emissions is highlighted separately. Source data are provided as a source data file.

| Table 4 Production data [$q_{b,n,i,conv}$] for food-specific products and share in broad and narrow categories for 2016 in Germany. | | | | | |
|---|---|---|---|---|---|
| **Production data** | | | | | |
| **Broad categories [b]** | **Share in broad categories** | **Narrow categories [n]** | **Share in narrow categories** | **Food-specific [i]** | **Total production quantity (in 1000 t) [$q_{b,n,i,conv}$]** |
| Plant-based | 7% | Vegetables | 98% | Field vegetables | 3166 |
| | | | 2% | Tomatoes | 78 |
| | | | | Other | 63 |
| | 2% | Fruit | 100% | Fruit | 1183 |
| | | | | Other | 0 |
| | 33% | Cereal | 5% | Rye | 733 |
| | | | 82% | Wheat | 13,026 |
| | | | 1% | Oat | 101 |
| | | | 13% | Barley | 2080 |
| | | | | Other | 0 |
| | 54% | Root Crops | 100% | Potatoes | 8577 |
| | | | | Other | 17,800 |
| | 1% | Legumes | 100% | Beans | 148 |
| | | | | Other | 280 |
| | 3% | Oilseed | 100% | Rapeseed | 1595 |
| | | | | Other | 61 |
| Animal based | 8% | Eggs | 100% | Eggs | 716 |
| | | | | Other | 0 |
| | 17% | Poultry | 100% | Broilers | 1510 |
| | | | | Other | 0 |
| | 13% | Ruminants | 100% | Beef | 1098 |
| | | | | Other | 18 |
| | 62% | Pork | 100% | Pork | 5559 |
| | | | | Other | 0 |
| Dairy | 100% | Milk | 100% | Milk | 31,736 |
| | | | | Other | 0 |

Production data were obtained from the German Federal Office of Statistics[88] and AMI[89,90].
Source data are provided as a source data file.

However, the choice of the farming system shows a much stronger effect when it comes to percentage surcharges (Table 3 and Fig. 2). This is due to the fact that the producer price of organic food is consistently higher compared to conventional food. Absolute external costs lead to a less significant percentage price increase for organic products emphasizing the difference between these two production types. Conventional animal-based products would require the highest relative percentage price

increase (146%), whereas organic plant-based products would require the lowest (6%) of all broad categories.

## Discussion

In the following, the emission differences between food categories and production methods as well as the internalization of external costs itself will be discussed.

As the results show, the production of animal-based products —especially of meat—causes the highest emissions. These results are in line with the prevailing scientific literature[12–15,48]. Such high emissions stem from the resource intensive production of meat, because of an inefficient conversion of feed to animal-based products. For beef cattle, this conversion ratio is reported by Pimentel and Pimentel to be as high as 43:1, meaning that 43 kg of feed are needed to produce 1 kg of beef product. These ratios differ significantly within meat categories, with broilers having the lowest ratio of all meat with only 2.3:1[49]. Furthermore, emissions from the animal itself through manure and digestion, as well as heating of stables, are also relevant factors which contribute to the high emissions of animal-based products. Secondary animal-based products, such as milk and eggs, however, cause lower emissions than meat. Again, these findings are in line with other sources[15,50]. This can be derived from the fact that the mass of milk or eggs a farm animal produces during its life is significantly higher than its own body weight on the day of slaughter. Thus, the same amount of resource input leads to a significantly higher amount of secondary (eggs, milk, etc.) than primary (meat) animal-based products. Hence, emissions from these resource inputs have a far smaller weight in secondary animal-based products.

Looking at the emission differences between conventional and organic production, the lower emissions of organic products in all three broad categories can be explained by the stricter rules under which organic farming is practiced. The EU-Eco regulation (2013) prohibits the use of mineral nitrogen fertilizers on organic farms. Therefore, direct emissions from the soil on which the fertilizer is used, and indirect emissions due to fertilizer production are lower compared to conventional production. Although the question to which extent animal manure causes less $N_2O$ emissions than nitrogen fertilizers in the form of direct soil emissions is controversial[51], a more careful nutrient handling on organic farms poses further explanation as to why considerable direct $N_2O$ emissions are avoided on said farms[52]. With regard to the feeding of animals (emissions of which are always allocated to the respective animal-based products in this study; cf. "Method and data" subsection on input data) on an organic farm, Article 14d of the EU-Eco regulation stipulates that only organic feed— mainly produced on the local farm (or other organic farms from the same region)—may be used. As our results in the subsection on quantification show, organically produced plants emit less GHG compared to their conventional counterparts. This notion can also be translated for the production of fodder plants. GHG emissions are thus saved by the more climate-friendly cultivation of organic fodder. Longer transport routes are also avoided as organic practice largely prohibits the use of imported fodder, which in the case of conventional agriculture in Germany includes rapeseed meal and maize from mostly Russia and Ukraine as well as soy from Brazil and Argentina. The cultivation of soy in these countries is associated with significant LUC emissions, which consequently are not applicable to organic products. The feed of organic dairy cows incorporates a significantly higher proportion of grazing (29.5% compared to 0.5%), which also avoids GHG emissions associated with the production of industrial feed for conventional dairy cows[53]. Moreover, the use of grassland instead of farmland leads to the preservation of $CO_2$ sinks[54]. However, the difference between farming practices is lower in both primary, and secondary animal-based products compared to the difference in plant farming. This may be explained with the higher use of land due to organic regulations prescribing a certain amount of land per animal, which is higher compared to average conventional production[22–24], as well as a higher living age and lower productivity of organically produced feed and raised animals[53] (cf. Table 2). This counterbalances or even reverses the described positive aspects of organic animal farming. Latter is the case for the narrow categories eggs, poultry as well as ruminants, for which organic farming results in higher emissions. For pork, however, organic farming achieves lower emissions. Such divergence of the ratio between farming system's emissions inside the animal-based category is explained by the different input quantities of soymeal (and the associated LUC emissions) into each product. As LUC emissions constitute a large share of the total emissions of a conventional animal-based product, the disbenefit of conventional products mainly depends on how large this share is. As this share is highest for pork (72%), it is the only subcategory of animal-based products, where organic farming results in lower emissions per kg. However, as the emissions of pork and their external costs are weighted the strongest inside the animal-based category (due to their high production quantity), the emission advantage of organic farming is passed on to the results for the broad category of animal-based products.

Further doubt toward a transition to organic farming was spread by Smith et al.[55], who rightfully addressed the potential increase of emissions resulting from a complete transition from conventional toward organic farming, given consumption patterns stay the same. These increases are thought to result from a higher amount of imported food, due to lower (regional) yields from organic farming. The financial incentives of internalization presented in our paper and the associated changing consumption patterns, however, pose a solution to these identified problems. Due to price elasticities of demand for food products (which are consistently regarded as normal goods in economic literature), appropriate pricing of food would make products of organic production more competitive compared to their conventional counterparts[56]: customers would increasingly opt for organic foodstuff due to the lowered price-gap between the two options. Although organic products are not always associated with lower emissions than conventional products (in the case of eggs, poultry, and ruminants), percentage price increases of organic products are consistently lower than for conventional products. Correspondingly, decreases in demand are lower for organic products. Thus, there would be a consistent advantage for organic products along with all products categories. This could potentially press the boundaries of land use for agriculture as organic practices mostly require more land than conventional systems due to lower yields[57–59]. However, our results suggest an increase in the prices of animal-based products to a significantly larger extent than the prices of plant-based products. The presumed consequential decline of animal-based product consumption would free an enormous landmass currently used for feed production. Further expansion of area-intensive organic agriculture would subsequently be made possible[60]. Furthermore, there is evidence that a shift from conventional to organic practices would indeed be beneficial for the ecosystem services and long-term efficiency provided by the particular land area[1,61]. If one takes into account the temporal change in yield difference which would result by converting farms from conventional to organic farming, there is scientific consensus that the yield gap will decrease over time[62,63]. Comparative studies between different cultivation methods also show that organic farming has lower soil-borne GHG emissions and higher rates of carbon sequestration in the

soil[52,64]. Soil degradation resulting from conventional systems would slow down or could even be reversed by changing to organic farming[19,65].

The internalization of external costs would also likely result in a lowered amount of thrown away food as appreciation for food would rise with its increased monetary value[66]. Thereby, further positive effects on efficiency and the environmental burden of food production would be achieved. Furthermore, a change in demand toward low-carbon (organic plant-based) food products is shown by Springmann et al. to positively affect the well-being and health of the individual, whereby national spending in health care could be reduced[67].

Price surcharges for externalities might be perceived as an additional financial burden for consumers[68]. It must be considered, however, that the costs of today's agricultural externalities are paid for by society and thus also by the individual already. This is yet done indirectly, for example, through emergency aid payments for floods or droughts and other increasing extreme weather conditions as an effect of global warming. When external costs are internalized, however, it would be possible for these external costs to be paid according to the polluter-pays principle[6] and thereby in an arguably fairer way. Following this principle, consumers demanding environmentally detrimental foodstuff would directly pay for its damages, whereas environmentally conscious consumers not wishing to support unsustainable farming practices are not financially burdened with its implications.

There is an opportunity to avoid or mitigate future damage by using additional government revenues resulting from the internalization of external climate costs: a subsidy policy providing greater incentives for sustainable agriculture at the farm level could be established. This could be done by ensuring that all received money from internalization is redistributed. Redistribution, which is the responsibility of national and international economic policy, should be carried out in particular for the benefit of the farmers concerned and should incentivize them to reduce their environmental impact. At the same time, social compensation appears to be necessary in order to help economically disadvantaged citizens, who are spending a far higher proportion of their income on food than economically more privileged groups. Surely, there are many political controversies implied in internalization policies. A thorough discussion of them, however, shall not be elaborated here in greater detail, since this paper's main focus is to deliver the quantitative basis for such political discourse.

This paper laid out a method to calculate product-specific external costs in the context of GHG emissions for foodstuff from German agricultural production. There is wide-ranging applicability of the method presented here. It can, for example, be used to assess the costs of further externalities, as databases such as the used GEMIS offer further data (such as externalities concerning nitrogen discharge or energy consumption), not only for Germany but also other regional contexts. We present many entry points from which to draw upon and add to the evolving literature on the true costs of food. Furthermore, a concern for current LCA methods, and thus a highly relevant research area, is the question of how to implement LUC emissions on a product-specific level. Since the focus of this study is on German production, LUC emissions are of negligible proportion for locally grown products, as agricultural land area is slightly decreasing in Germany[55]. For animal-based products, however, a significant amount of emissions arise due to additional LUC emissions from feed imports. We calculate such emissions with the method of Ponsioen and Blonk[45], whereby the shortcomings of common direct and indirect LUC assessment are largely prevented, and emissions are calculated on the basis of available statistical land-use data for a specific country. However, as there currently are different scientific approaches to LUC assessment, we list LUC emissions separately from other types of emissions. The here analyzed stage of agricultural production, assessed within the system boundaries of cradle to farmgate, causes the greatest externalities along the value chain of foodstuff[69]. Despite this, further research should also be conducted for the activities succeeding the farmgate (e.g., processing and logistics) and corresponding externalities.

The approach presented here represents a contribution to the true costs of food, which—even with partial implementation—could lead to an increase in the welfare of society as a whole by reducing current market imperfections and their resulting negative ecological and social impacts.

## Methods

**Outline**. In this section, first, we outline the method as a whole to give the reader an orientation and context for the following two parts. Second, we discuss the input data (for quantification and monetization). Third, we explain the merging of all input data, and thus the calculation of the output data. Finally, we address the influence of uncertainties on our method. The reference year for this analysis is 2016, and the reference country is Germany, which is listed as the third most affected country in the Global Climate Risk Index 2020 Ranking[70].

**Method in short**. We differentiate between two steps within this method of calculating food-category-specific externalities and the resulting external costs. These are first the quantification and second the monetization of externalities from GHGs (visualized in Fig. 3). We use this bottom-up approach following the example of Grinsven et al.[10], who conducted a cost-benefit analysis of reactive nitrogen emissions from the agricultural sector. This two-stepped method also allows the adequately differentiated assessment for GHG emissions of various food categories.

The quantification includes the determination of food-specific GHG emissions —also known as carbon footprints[39]—occurring from cradle to farmgate by the usage of a material-flow analysis tool. Carbon footprints are understood within this paper in line with Pandey et al.[71] where all climate-relevant gases, which (in addition to $CO_2$) include methane ($CH_4$) and nitrous oxide ($N_2O$), are considered. Their 100-year $CO_2$ equivalents conversion factors are henceforth defined as 28 and 265, respectively[72]. Here, the material-flow analysis tool GEMIS (Global Emission model for Integrated Systems)[44] is used, which offers data for a variety of conventionally farmed foodstuff. As GEMIS data focus on emissions from conventional agricultural systems, we carried out the distinction to organic systems ourselves. We determined the difference in GHG emissions between the systems by applying meta-analytical methods to studies comparing the systems' GHG emissions directly to one another. Meta-analysis is commonly used in the agricultural context, for example, when comparing the productivity of both systems[57–59] or their performance[1].

For better communicability, we first aggregate the 11 food-specific datasets given in GEMIS to the broader food categories plant-based, animal-based, and dairy by weighting them with their German production quantities (cf. "Results" subsection on quantification). On top of that, LUC emissions are calculated for conventional foodstuff.

Through monetization, these emission data are translated into monetary values, which constitute the category-specific external costs. The ratio of external costs to the foodstuff's producer price represents the percentage which would have to be added on top of the current food price to internalize externalities from GHGs and depict the true value of the examined foodstuff.

**Input data for quantification**. Starting with the data on food-specific emissions, GEMIS is used because of its large database of life-cycle data on agricultural products with a geographic focus on Germany. GEMIS is a World-Bank acknowledged tool for their platform on climate-smart planning and drew on 671 references, which are traced back to 13 different databases. The German Federal Environmental Agency uses GEMIS as a database for their projects and reports establishing it to be an adequate tool for the German context especially[73,74]. This tool is provided by the International Institute for Sustainability Analysis and Strategy (IINAS). GEMIS offers a complete view on the life cycle of a product, from primary energy and resource extraction to the construction and usage of facilities and transport systems. As GEMIS only offers data for the year 2010, we conducted a linear regression on the basis of the prevailing emission trend for the German agricultural context in order to align the data with the reference year 2016[75]. For this, annual German emission data from 2000 to 2015 from the Federal Environmental Agency of Germany was used[76]. On every level of the process chain, data on energy- and material-input, as well as data on output of waste material and emissions, are provided by GEMIS. These data consist partly of self-compiled data from IINAS and partly of data from third-party academic research or other life-cycle assessment tools. Specific information on the data sources is available for

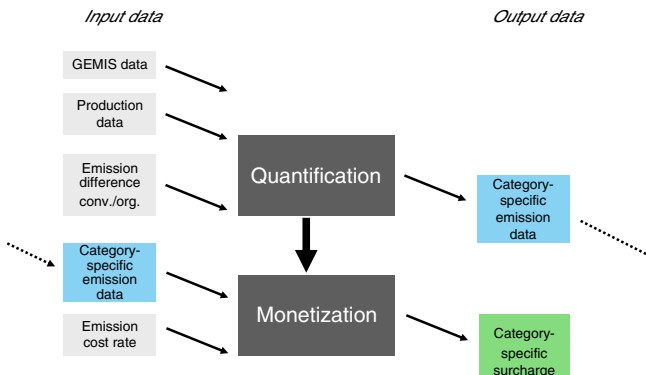

**Fig. 3 Visualization of the method.** The method includes quantifying and monetizing product-specific externalities. In the case of Germany, emission data were obtained from the Global Emission Model for Integrated Systems (GEMIS)[44]. We used production data from the German Federal Statistical Office[88] and AMI[89,90], and calculated the emission difference between organic and conventional production based on a meta-analytical approach (see "Results" subsection on input data for quantification). The category-specific emission data were calculated on the basis of these input data. The emission cost rate was obtained from the German Federal Environmental Agency (UBA)[32]. The category-specific external costs were determined on the basis of the previously developed price-quantity-framework (see "Results" subsection on input data for monetization).

every dataset of a product. In this study, the system boundaries for assessing food-specific GHG emissions span from cradle to farmgate. This means that we consider all resource inputs and outputs during production up to the point of selling by the primary producer (farmgate). This includes emissions from all production-relevant transports as well as emissions linked to the preliminary building of production-relevant infrastructure.

We specify that for animal-based products, emissions from feed production, as a necessary resource input, are assigned to these animal-based products. Such emissions naturally should include LUC emissions. LUC emissions are of negligible proportion for locally grown products, as agricultural land area is slightly decreasing in Germany[47]. Thus, we have to focus solely on imported feed for conventional animal-based and dairy products. Organic feed is not considered as article 14d of the EU-Eco regulation stipulates that organic farms have to primarily use feed which they produce themselves or which was produced from other organic farms in the same region[77]. The region is understood as the same or the directly neighboring federal state[46]. Although the EU-Eco regulation does not completely rule out fodder imports from foreign countries, it limits its application significantly. Also, one has to consider that over 60% of the organic agricultural area belongs to organic farming associations[78]. These associations stipulate even stricter rules than the standard EU-eco regulation. Examples are Bioland, where imports from other EU and third countries are only allowed as a time-limited exception[79], Naturland, where additionally imports of soy are banned completely[80], or Neuland, that ban any fodder imports from overseas[81]. We thus assume that the emissions that could possibly be caused by organic farming in Germany through the import of feed constitute a negligibly small fraction of the total emissions of a product. Thus, we follow common assumptions from the literature[82–84] and calculate no LUC emissions for organic products. For conventional products, we calculate LUC emissions by application of the method of Ponsioen and Blonk[45]. This method allows the calculation of LUC emissions for a specific crop in a specific country for a specific year. With regards to the year, we apply our reference year 2016. With regards to crop and country one has to keep in mind that in the case of Germany, the net imports of feed are the highest for soymeal, followed by maize and rapeseed meal, making up over 90% of all net positive feed imports[85]. Maize and rapeseed meal are both imported mainly from Russia and Ukraine (93% and 87% of all imports[86]). Taken together, the crop area of Russia and Ukraine is decreasing by 150,000 ha/year (data from 1990 to 2015 were used[87]). Following Ponsioen and Blonk[45], we thus assume that there are no LUC emissions of agricultural products from these countries. This leaves us with soymeal, of which 97% are imported from Argentina and Brazil. We thus calculate LUC emissions of soymeal for Argentina and Brazil, respectively. Data are used from Ponsioen and Blonk[45], except for the data of the crop area, where updated data from FAOSTAT are used in order to match the reference year. We then weigh those country-specific emission values according to their import quantity. This results in 2.54 kg $CO_2$eq/kg soymeal. To incorporate this value into the conventional emission data from GEMIS, we map the LUC emissions to all the soymeal inputs connected to the food-specific products.

For aggregation to narrow categories, we categorize every dataset from GEMIS into one of the eleven narrow food categories. The choice of separation into these specific categories is based on the categorization of the German Federal Office of Statistics[88] from which production data were obtained. According to one category's yearly production quantity, we incorporate every food product into the weighted mean of its corresponding food category. Thus, the higher a food's production quantity, the greater the weight of this product's emission data in the broad category's emission mean. All data on the production quantities refer to food produced in Germany in the year 2016. For this weighting and aggregation step, only production quantities used for human nutrition were considered, thus feed and industry usage of food are ruled out (in contrast to emission calculation, where feed is indeed considered). Besides the German Federal Office of Statistics[88], the source for this data is the German Society for Information on the Agricultural Market (AMI)[89,90]. Only production data for conventional production is used. Thereby, we imply ratios of production quantities across the food categories for organic production that are equal to those of conventional production. This does not fully reflect the current situation of organic production properties but allows for a fair comparison between the emission data of organic and conventional food categories. Doing otherwise would create ratios between emission values of organic and conventional broad categories that would not be representative of the ratios between organic and conventional narrow categories. In Table 4, all production data are listed, whereby total production quantities in 1000 t can be found in the right column. Translating these into percentage shares, the column right to the narrow category's column represents the shares of the specific foods inside the narrow categories, whereas the column right to the broad category's column represents the shares of the narrow categories inside the broad categories. These shares are expressed in formula 2a and 2b (see "Method and data" subsection on output data) by the terms $\frac{p_{b,n,conv}}{P_{b,conv}}$ (share in broad categories) and $\frac{q_{b,n,i,conv}}{p_{b,n,conv}}$ (share in narrow categories).

We aggregate GEMIS emission data ($q_{b,n,i,conv}$) to narrow ($e_{b,n,conv}$) and broad categories ($E_{b,conv}$) by multiplying the respective emission data with the shares from Table 3 (cf. formula 2a and b, "Method and data" subsection on output data). From these conventional emission values, we derive emissions for organic production. For narrow as well as broad categories, the respective conventional emission values are multiplied with the applicable emission differences $D_{b,org/conv}$ (cf. Table 2).

With these data, we aggregate the above mentioned eleven food categories to three broad categories: plant-based, animal-based, and dairy. Besides the obvious differentiation between animal- and plant-based products, dairy is considered separately from other animal-based products because of its relatively high production volume and its, in contrast to that, relatively low externalities. Because the weighted mean of the three main categories is affected by the production quantities of its corresponding subcategories, mapping dairy into the animal-based category would otherwise distort the emission data of this very category.

As outlined before, only data regarding externalities of conventional agricultural production are included in GEMIS and could therefore be aggregated. Nevertheless, by applying meta-analytical methods regarding the percentage difference of GHG emissions between conventional and organic production, we derive the emission data for organic production for each of the broad categories (plant-based, animal-based, and dairy). It has to be noted that LUC emissions are consistently excluded at this level of calculation. To derive emission differences between organic and conventional farming, research was conducted by snowball sampling from already existing and thematically fitting meta-analysis, by keyword searching in research databases, as well as forward and backward search on the basis of already-known sources. Criteria for selected studies were climatic and regulative comparability to Germany. In the selected studies, relative externalities between conventional and organic farming are compared in relation to the cropland. To cover a reasonably relevant period, we decided to search for studies published within the past 50 years (from 1969 to 2018) and could therefore identify fifteen relevant studies, spanning from 1995 to 2015. Four of these studies have Germany as their reference country while the other eleven focus on other European countries (Denmark, France, Ireland, Netherlands, Spain, UK; please consult Table 2 for specifics). The weighted mean of the individual study results amounts to the difference in GHG emissions between the two farming production systems. As the selected studies are based on geophysical measurements and not on inferential statistics, a weighting based on the standard error of the primary study results like in standard meta-analysis[91] was not possible. We aimed for a system that weights the underlying studies regarding their quality and therefore including their results weighted accordingly in our calculations. Within the scope of classic meta-analyses[92], the studies' individual quality is estimated according to their reported standard error (SE), which is understood as a measure of uncertainty: the smaller the SE, the higher the weight that is assigned to the regarding source. Due to the varying estimation methods of considered studies, the majority of considered papers does not report measures of deviation for their results. These state definite values; therefore, there is no information about the precision of the results at hand. Against this background, we have decided to use a modified approach to estimate the considered papers' qualities[93]. Following van Ewijk et al.[94] and Haase et al.[95], we apply three relevant context-sensitive variables to approximate the standard error of the dependent variable and thereby evaluate the quality of each publication: the newer the paper (compared to the timeframe between 1995 and 2018), the higher we assume the quality of reported results. The

more often a paper was cited per year (measured on the basis of Google Scholar), the higher the paper's reputation. The higher the publishing journal's impact factor (measured with the SciMago journal ranking), the higher its reputation and therefore, the paper's quality. For every paper, the three indicators publishing year (shortened with PY in Table 2), citations/year (CY), and journal rank (SJR) rank a paper's impact on a scale from 1 to 10, where 1 describes the lowest qualitative rank and 10 the highest. The sum of these three factors (SUM) then determines the weight of a paper's result in the mean value (WEIGHT). The papers' reported emission differences between organic and conventional (diff. org/conv) are weighted with the papers' specifically calculated WEIGHTS and finally aggregated to the emission difference between both systems.

With this approach, we weight results of qualitatively valuable papers higher and are therefore able to reduce the level of uncertainty in the estimated values because standard errors could—due to inconsistencies in the underlying studies—not be used. The results of this meta-analytical approach are listed in Table 2 (cf. "Results" subsection on quantification); further details can be found in Supplementary Note 1 and Supplementary Table 1. The studies considered compare GHG emissions of farming systems in relation to the crop/farm area. However, since our study aims to compare GHG emissions in relation to the weight of foodstuff, we include the difference in yield (yield gap) between the two farming systems for plant-based products and the difference in productivity (productivity gap) for animal-based and dairy products. For plant-based products, the yield gap is 117%, meaning that conventional farming produces 17% more plant-based products than organic farming in a given area. This gap was derived from three comprehensive meta studies[57–59] and weighted as just described for the emission difference between organic and conventional farming. For animal-based as well as dairy products, the productivity gap could be determined with the same studies used for the meta-analytical estimation of the emission differences[22–25,28,95]. The productivity gap is 179% for animal-based and 152% for dairy products. In line with Sanders and Hess[63], the yield (or productivity) difference $\frac{yield_{conv}}{yield_{org}}$ affects the calculation of the food-weight-specific emission difference $\frac{GHG_{org\ food\ weight}}{GHG_{conv\ food\ weight}} = D_{org/conv}$ between both farming systems: the yield difference is hereby multiplied with the cropland-specific emission difference $\frac{GHG_{org\ cropland}}{GHG_{conv\ cropland}}$. Resulting from this, the emission difference can be formulated as follows:

$$D_{org/conv} = \frac{GHG_{org\ food\ weight}}{GHG_{conv\ food\ weight}} = \frac{GHG_{org\ cropland}}{GHG_{conv\ cropland}} \times \frac{yield_{conv}}{yield_{org}} \qquad (1)$$

If the yield difference were not included, emissions from organic farming would appear lower than they actually are as organic farming has lower emissions per kg of foodstuff but also lower yields per area. With formula 1, we adjust for that.

**Input data for monetization.** Monetization of these externalities requires data on GHG costs as well as data on the food categories' producer prices.

The cost rate for $CO_2$ equivalents used in this study stems from the guidelines of the German Federal Environment Agency (UBA) on estimating external ecological costs[32]. They recommend a cost rate of 180 € per ton of $CO_2$ equivalents.

This value is very close to the value of the 5th IPCC Assessment Report (173.5 €/tCO₂eq), where the mean of all (up to this point) available studies with a time preference rate of 1% was determined[33]. The cost rate from the German Federal Environment Agency's guideline is based on the cost damage model FUND[96] and includes an equity weighting as well as a time preference rate of 1% for future damages. In this model, different impact categories are considered in order to estimate external costs from GHG emissions. Damage costs can be differentiated as benefit losses such as lowered life expectancy or agricultural yield losses and costs of damage reduction such as medical treatment costs or water purification costs[97]. Following UBA, these damage costs are analyzed in the following categories: agriculture, forestry, sea-level rise, cardiovascular and respiratory disorders related to cold and heat stress, malaria, dengue fever, schistosomiasis, diarrhea, energy consumption, water resources, and unmanaged ecosystems[96]. Using a cost-benefit-analysis (CBA), an adequate level of emissions is reached when marginal abatement costs are equal with damage costs. In a CBA external damage, costs can therefore be conceptualized as a price surcharge necessary to effect their optimal reduction[98].

For the pricing of the food categories, we determine the total amount of proceeds that farmers accumulate for their sold foodstuff in €[99] for each category (producer price) divided by its total production quantity. Thereby we calculate the relative price per ton for each foodstuff. We solely refer to producer prices as the system boundaries only reach until the farmgate.

**Calculating output data.** Output data include the aggregation and separation of food-specific categories to the broader categories of animal- and plant-based products, as well as conventional and organic products. As previously explained, such aggregation and separation are needed because the underlying material-flow analysis tool only lists food-specific emission data for conventionally produced foodstuff. Combining the input data, we are now able to quantify and monetize externalities of GHGs for different food categories.

For quantification, we separate between the following two steps: first, the aggregation of emissions data to broader categories and second the differentiation between conventional and organic farming systems. We iterate these steps two times, once for broad categories of animal-based products, plant-based products, and dairy and once for more narrow categories of vegetables, fruits, root crops, legumes, cereal, and oilseeds on the plant-based side as well as milk, eggs, poultry, ruminant, and pig on the animal-based side. Figure 4 displays the whole process of quantification schematically before we describe it in detail in the following text.

Concerning the reasoning behind the method, the question that might come to mind is why the differentiation between farming systems happens after the aggregation and not before. This is due to the fact that the proportional production quantities of specific food as well as food categories to each other differ from conventional to organic production. Let us imagine aggregation would take place after the differentiation of farming systems: for example, beef actually makes up over 50% of all produced food in the organic animal-based product category, while it only accounts for 25% of the conventional animal-based product category (cf. production values in Table 3). As beef production produces the highest emissions of all foodstuffs, these high emissions would be weighted far stronger in the organic category than in the conventional category and thereby producing a higher mean for the organic animal-based product category than for the conventional one. As

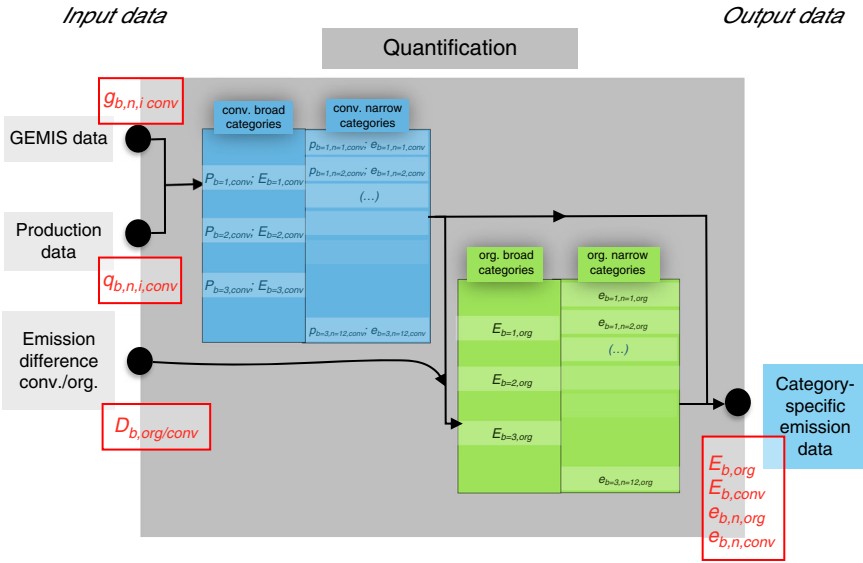

**Fig. 4 Visualization of the quantification process.** Quantification as well as corresponding input and output data are displayed. Data from the Global Emissions Model for Integrated Systems (GEMIS)[44] ($g_{b,n,i,conv}$) and production data[88–90] ($q_{b,n,i,conv}$) are combined, and emission data for broad ($E_{b,conv}$) and narrow ($e_{b,n,conv}$) categories are derived for conventional production. Organic emission values are calculated by multiplication of conventional emission values ($E_{b,org}$ and $e_{b,n,org}$) with the emission difference ($D_{b,org/conv}$) (cf. "Input data for quantification").

can be seen from this example, the organic animal-based product category could have a higher mean of emissions than the conventional animal-based product category while still having lower emissions for each individual organic animal-based product than conventional production. Deriving GHG emissions of foodstuff before aggregating to broader categories would thus be problematic and create means not representative for the elements that make up the broader category. To prevent this problem, the chosen method in this paper is thus to first aggregate to the chosen level of granularity (broad or narrow food categories) and then to derive emissions of organic production from conventional production data.

The first step of aggregation consists first of aggregating food-specific emission data from GEMIS $g_{b,n,i,conv}$ to the narrow categories $e_{b,n,conv}$ and second aggregating emission data from the narrow categories to the broad categories $E_{b,conv}$. As mentioned before and remarked in the respective indices, all these data only refer to conventional production up to this point. For both steps, the method is identical. The aggregation to narrow categories is represented in (2a) where $e_{b,n,conv}$ stands for the emissions of the narrow category $n$, which itself is part of the broad category b. Input data from GEMIS are remarked as $g_{b,n,i,conv}$, whereby the index $i$ refers to the $i$th element of category $n$. It's production quantity is $q_{b,n,i,conv}$. $p_{b,n,conv}$ represents the production quantity of the narrow category $n$. $I$ (and $N$ in formula 2b) represents the highest index of an element in a narrow (or a broad) category.

$$e_{b,n=x,conv} = \sum_{i \in n=x}^{I} g_{b,n,i,conv} \times \frac{q_{b,n,i,conv}}{p_{b,n,conv}} \tag{2a}$$

The aggregation to broad categories is described by formula 2b whereby $E_{b,conv}$ are the emissions and $P_{b,conv}$ the production quantity of broad category b.

$$E_{b=x,conv} = \sum_{n \in b=x}^{N} e_{b,n,conv} \times \frac{p_{b,n,conv}}{P_{b,conv}}. \tag{2b}$$

In the second step, we calculate emission values for organic production by multiplying the calculated emission difference $D_{b,org/conv}$ between both farming systems (cf. "Input data for quantification") with the conventional emission values. These organic emission values are denoted as $E_{b,org}$ for broad categories and $e_{b,n,org}$ for narrow categories.

To calculate the costs $C_b$ of category-specific emissions, we multiply the cost rate $P$ for $CO_2$ equivalents with the category-specific emission data $E_b$ or $e_{b,n}$ (depending on whether broad or narrow categories are observed). Further, we determine percentage surcharge costs $\Delta_b$ by setting these costs in relation to the producer price $pp_b$ of the respective food category: $\Delta_b = \frac{C_b}{pp_b}$ (the calculation is analogue for narrow categories). These surcharge costs represent the price increase necessary to internalize all externalities from GHG emissions for a specific food category.

**Dealing with uncertainties**. Due to the interdisciplinarity and novelty of our study, we connect several methodological approaches and refer to various sources for data. Against this background, we had to accept some uncertainties while assembling and using the developed framework for our calculation. The studies included in our meta-analytical approach of calculating the difference between organic and conventional emission values, for one, are not fully consistent in the methodologies each of them uses (refer to Supplementary Table 1 for details). Furthermore, from the results of all included studies, it is apparent that there exists a wide range of emission differences between the farming practices, depending on the paper's scope and examined produce[21]. We attempted to account for this by performing the studies according to their fit regarding the object of research (cf. "Input data for quantification"). Due to insufficient availability of the data for the emission differences between organic and conventional on the basis of each narrow category, an average for the emission difference was used. This possibly results in imprecisions during the internalization of the external costs on the level of all narrow categories. Therefore, we focus on the aggregated broad categories, as this uncertainty can be evaded here. Furthermore, the in literature reported price factor for $CO_2$ equivalents is volatile over time, impacting the results of this paper. It is to be expected that the external costs of GHG emissions are likely to rise in the future (cf. subsection on research aim and literature review). Also, our study's scope is confined to the assessment of the current production situation within the German agricultural sector. Therefore, we do not account for future developments regarding a changing agricultural production landscape after internalization of the accounted external costs. We do, however, discuss possible effects on demand patterns as well as the environmental and social performance of the agricultural sector in "Discussion". Regarding the incorporated LUC emissions, there appears to be a lacking scientific consensus on a general method of calculation for such emissions[45,100–102]. We thus want to emphasize that these additional emissions should be treated with caution and are thereby displayed separately from the other data.

**Reporting summary**. Further information on research design is available in the Nature Research Reporting Summary linked to this article.

## Data availability

The datasets generated and analyzed during the current study are available in the Center for Open Science repository, https://osf.io/e7v8x/?view_only=0bff6aa858a340df9046816c1404a51c. The datasets are derived from the following databases: German Federal Office of Statistics (https://www-genesis.destatis.de/genesis/online), German Society for Information on the Agricultural Market (AMI) (https://www.ami-informiert.de/), KTBL-Standard Gross Margins (https://daten.ktbl.de), EU Open Data Portal (https://data.europa.eu/euodp/en/data/dataset/uLrJZE2PQkMHod6feE8gXQ), Eurostat (https://ec.europa.eu/eurostat/databrowser), German Federal Office for Agriculture and Food (BLE) (https://www.ble.de), German Head Organization of Ecological Food Economics (BÖLW) (https://www.boelw.de/), Expert Agency for Renewable Resources (FNR) (https://fnr.de/), and the German Federal Ministry for Food and Agriculture (BMEL) (https://www.bmel-statistik.de/). More detailed information is provided in the source data file. Microsoft Excel (for Mac, version 16.16.26) was used to calculate and analyze the data of this study. Emission values were derived from the publicly available material-flow analysis tool GEMIS (Version 4.95), which can be downloaded here: http://iinas.org/gemis-download-121.html. Source data are provided with this paper.

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

## Acknowledgements

We want to thank Joe F. Bozeman III (University of Illinois at Chicago), Jules Pretty (University of Essex), and Till Weidner (University of Oxford) for their valuable support during the revision phase of this paper. Their comments and insights helped to improve the quality of this paper.

## Author contributions

T.G. supervised the project. All authors developed the concept and designed the framework. M.P. and A.M. gathered the data. All authors discussed the results and implications. M.P. wrote the paper. All authors contributed equally and extensively to the revision process.

## Funding

## Competing interests

The authors declare no competing interests.
