## [Peer Review File · Nature Communications]

Reviewers' comments:

Reviewer #1 (Remarks to the Author):

The manuscript by Pieper et al., "How much is the dish?" presents a generalised approach to internalise the climatic externalities caused by conventional and organic food production systems, primarily focussing on Germany. They use a material flow analysis tool GEMIS to estimate the GHG emissions during the conventional production of different food categories (broadly categorised into plant-based, animal-based and milk) with 11 subcategories, that occur until the farm gate. To estimate the emissions associated with organic farming, the authors mention that they estimate these from the meta-analysis of the suitable literature. The study found that organic produce has lower environmental externality in all the cases compared to their conventional counterparts and plant-based organic food has the lowest environmental externalities. Adding these costs to the markup price will raise the prices of conventionally produced food to a larger extent, potentially making the organic food (which is also found to be more environment-friendly) more competitive in the market. The study topic is relevant in the present context, where it is very important that we find effective methods to include environmental externalities as a part of the true cost of production. The manuscript, however, does not provide several fundamental information on the approach and misses some crucial factors in their assumptions. Overall, due to these, I am not fully convinced about the values of externalities and the true costs estimated by the authors and their application to German food production. I will detail these below. In addition, I also find the language to be repetitive, unnecessarily descriptive and following not so standard formats. I explain the specific points below.

I. Comments on the approach.

1. Authors mention that they used meta-analysis to estimate the difference between the conventional and organic farming. The studies they used are listed in Table 2 and it is the only place where authors inform about the literature they used to estimate emissions from organic farming. These papers are missing in the reference list.

Authors also do not mention the key emission values they found in the literature, how they were estimated (e.g. empirical data or models), regional coverage, to what degree the studies covered the broader and specific food categories that the authors used, and whether the studies were consistent with the boundaries (i.e. the direct and indirect emissions) selected by the authors. I find it surprising that authors describe how they ranked the literature based on citations and journal impact factor, but nowhere talk about the main information they used.

2. It is also not clear, how authors estimated the emissions from organic farming based on these studies, and what was the level of uncertainty in the estimated values. I would like to see this description very clearly because it is crucial information on which the study is founded.

3. Authors have cited several of their previous works on which the current study is built upon. Unfortunately, it was very difficult to assess those studies. Particularly the reference 16 is only an abstract accepted for a conference. It makes it very challenging to know how the current approach is built upon the author's previous work and compare them.

4. It is not clear whether authors include emissions associated with the transport of inputs such as fertilizers.

5. As authors already mention that yield in organic production is lower than the conventional, it may be argued that more land area will need to be cultivated for the same amount of production. However, this study does not cover the emissions resulting from the land-use conversion that will be required for this. I understand that incorporation of land-use change may complicate the study. At the same time, even if we assume that no LUC occurs in Germany, I may argue that the import of food will

increase. Given that emissions that occur outside Germany are excluded in the study, it might be underestimating the emissions.

6. The authors do not mention whether they take into account the temporal changes in emissions that can occur as the organic farming is followed on a continuous basis. It is very likely that the yield gaps between the conventional and organic farming may increase or decrease, and the same can happen with the soil-borne GHG emissions and soil carbon sequestration rates. Ideally, authors should account for uncertainties arising due to these, but if it is not possible due to the limitation of the data, these effects need to be discussed as a part of the uncertainty.

II. Format and language: 1. the format is not according to the journal, (methods are before the results). I assume that the editing team will take care of it.

2. Authors use words like 'external effects' and external costs, both of which I understand mean externality? I will suggest using the same term throughout to make it easy for the reader.

3. Page 3: what is chapter 3.2.1? I would prefer using 'section' instead of 'chapter'.

4. Page 3 paragraph 2, authors say that it will be technically incorrect to refer to the current approach as LCA and that the term carbon footprint is more appropriate. I, therefore, suggest authors talk about carbon footprint from the beginning.

5. page 8: Authors mention that they use 11 food categories but name only 10.

6. Table 1, 2 and 3 need reformatting. Units are at the bottom instead of being at the top. What is meant by field vegetables in table 1?

Primarily because of the missing crucial information on the methods used in the study along with the additional facts mentioned above, I will not recommend the article for publication.

Reviewer #2 (Remarks to the Author):

About the paper with the title ""How much is the dish?" – Calculating External Climate Costs for Different Food Categories" I have the following comments:

- I suggest the Authors to rewrite the abstract with a clear presentation of the main motivations, objectives, methodologies and main insights (avoiding in the abstract terms as mark-up, maybe not a familiar term for the general readers of these topics).

- The literature review is weak. I suggest improve and update significantly this part.

- In the current section 3 you need to support your options with other scientific papers already published. For example "...quantification and second the monetization of external effects from GHGs, visualized in...". Why this approach and not other?. Other example, "...analysis tool GEMIS (Global Emission-model for Integrated Systems) 25 is used which offers...". There are other approaches. Why not others? On other words, this section is too much descriptive in a part where you make several options.

- On the other hand, the Authors need to be more specific in this section. For example, "...individual countries was accessible, EU-data is used.". What this mean?

- Yet in the section 3, sometimes, it is hard to understand what part of the data was obtained by you and what part was obtained from other sources. For example, only in the subsection 3.2.1. we understand that "...basis of quantity- and emission-trend was conducted in order to align the data

with the...". How you did this? And the scientific support? In turn, sometimes, it is, also, hard understand the importance of each equation. For example, improve the explanations about the equation (1) and the relevance of this equation for your research. The same for the others. Another question is about the source of these equations that need to be clarified.

- In this section 4 you need to link more the results obtained with the equations and methodologies presented before and to compare more specifically these results with other works.

Response Sheet

Reviewer #1

I. Comments on the approach

1. A) “Authors mention that they used meta-analysis to estimate the difference between the conventional and organic farming. The studies they used are listed in Table 2 and it is the only place where authors inform about the literature they used to estimate emissions from organic farming. These papers are missing in the reference list.”

Response: Thank you very much for your attentive comment! We added a more detailed version of table 2 in the SI section, where all literature is referenced. The missing references shall also be listed here in occurring order (and with relating reference number) in the manuscript:

1. Tuomisto, H. L., Hodge, I. D., Riordan, P. & Macdonald, D. W. Comparing global warming potential, energy use and land use of organic, conventional and integrated winter wheat production. *Ann. Appl. Biol.* 161, 116–126 (2012).
2. Cooper, J. M., Butler, G. & Leifert, C. Life cycle analysis of greenhouse gas emissions from organic and conventional food production systems, with and without bio-energy options. *NJAS - Wagening. J. Life Sci.* 58, 185–192 (2011).
3. Aguilera, E., Guzmán, G. & Alonso, A. Greenhouse gas emissions from conventional and organic cropping systems in Spain. II. Fruit tree orchards. *Agron. Sustain. Dev.* 35, 725–737 (2015).
4. Aguilera, E., Guzmán, G. & Alonso, A. Greenhouse gas emissions from conventional and organic cropping systems in Spain. I. Herbaceous crops. *Agron. Sustain. Dev.* 35, 713–724 (2015).
5. Reitmayr, T. Entwicklung eines rechnergestützten Kennzahlensystems zur ökonomischen und ökologischen Beurteilung von agrarischen Bewirtschaftungsformen. (Buchedition Agrimedia, 1995).
6. Küstermann, B., Kainz, M. & Hülsbergen, K.-J. Modeling carbon cycles and estimation of greenhouse gas emissions from organic and conventional farming systems. *Renew. Agric. Food Syst.* 23, 38–52 (2008).
7. Casey, J. W. & Holden, N. Greenhouse Gas Emissions from Conventional, Agri-Environmental Scheme, and Organic Irish Suckler-Beef Units. *J. Environ. Qual.* 35, 231–9 (2006).
8. Flessa, H. et al. Integrated evaluation of greenhouse gas emissions (CO₂, CH₄, N₂O) from two farming systems in southern Germany. *Agric. Ecosyst. Environ.* 91, 175–189 (2002).
9. Basset-Mens, C. & van der Werf, H. M. G. Scenario-based environmental assessment of farming systems: the case of pig production in France. *Agric. Ecosyst. Environ.* 105, 127–144 (2005).
10. Dalgaard, R., Halberg, N., Kristensen, I. S. & Larsen, I. Modelling representative and coherent Danish farm types based on farm accountancy data for use in environmental assessments. *Agric. Ecosyst. Environ.* 117, 223–237 (2006).
11. Bos, J. F. F. P., Haan, J. J. de, Sukkel, W. & Schils, R. L. M. Comparing energy use and greenhouse gas emissions in organic and conventional farming systems in the Netherlands. in 439–442 (2007).
12. Thomassen, M. A., van Calker, K. J., Smits, M. C. J., Iepema, G. L. & de Boer, I. J. M. Life cycle assessment of conventional and organic milk production in the Netherlands. *Agric. Syst.* 96, 95–107 (2008).
13. Haas, G., Wetterich, F. & Köpke, U. Comparing intensive, extensified and organic grassland farming in southern Germany by process life cycle assessment. *Agric. Ecosyst. Environ.* 83, 43–53 (2001).

1. B) “Authors also do not mention the key emission values they found in the literature, how they were estimated (e.g. empirical data or models), regional coverage, to what degree the studies covered the broader and specific food categories that the authors used, and whether the studies were consistent with the boundaries (i.e. the direct and indirect emissions) selected by the authors.”

Response: Thank you for this very plausible notation. We have chosen the relevant papers according to our system’s boundaries, but it is of course important to make the comparability between used studies, their eligibility, as well as their stated emission values clear for the reader. Therefore, we have enhanced concerning table 2 with the information which the reviewer suggested. As the additional information made the aforementioned table significantly larger we suggest to put a slimmed down version of table 2 into the paper’s body; this version includes all crucial information for the further development and calculations. We suggest that the full table with all important information suggested by the reviewer will be available in the Supplementary Information and - for better reference - we added it in the following:

Supplementary Information, p. 46:

Source	Estimation method	Boundaries	Regional coverage ¹	Observed food category	Emission values [kgCO ₂ -eq/ha]		difference org/conv
					conventional	organic	
Plant-based							49%
Aguilera et al. (2011a) p. 719	LCA modelling (empiric data from interviews and other studies)	cradle to farmgate	Spain	cereals ²	1.024	361	45% ³
				legumes	568	232	
				field vegetables ⁴	3.448	1.418	
				vegetables greenhouse	11.841	7.592	
Aguilera et al. (2011b) p. 730	LCA modelling (empiric data from interviews and other studies)	cradle to farmgate	Spain	citrus fruits ⁵	6.324	1.897	49%
				fruits	2.597	1.480	
				wine	964	641	
Cooper et al. (2011) p. 189	empiric data gathered at site	direct and indirect emissions until farmgate; comparable with cradle to farmgate	Nafferton (Northern England), UK	crop rotation ⁶	2.019	841	42%
Küstermann et al. (2008) p. 48	modelling (software REPRO)	direct and indirect inputs until farmgate; comparable with cradle to farmgate	Scheyern (Upper Bavaria), Germany	crop rotation ^{7,8}	376	263	70%

¹ The specific regional coverage was not stated in all studies. Locations are stated as precisely as possible.

² We have excluded the in underlying study (Aguilera et al. 2011a) observed food category ‘rice’ for this assessment as it is an irrelevant product for the assessment of the German agricultural sector.

³ When there was more than one food category assessed in one study, we weighted them equally to not interfere with the weighting system between the studies.

⁴ In GEMIS ‘field vegetables’ constitutes a collective term describing vegetables that are grown in the open air. This form of cultivation is in contrast to the horticultural cultivation of vegetables which uses greenhouses, foil tunnels or other artificially protected areas.

⁵ We have excluded the in underlying study (Aguilera et al. 2011b)observed food categories ‘subtropical fruit trees’, ‘tree nuts’, and ‘olives’ as they are irrelevant products for the assessment of the German agricultural sector.

⁶ Rotation includes winter wheat, potatoes, beans, cabbage, and spring/winter barley.

⁷ Rotation includes potatoes, winter wheat, sunflower, winter rye, and maize.

⁸ Even if sunflower is irrelevant to the assessment of the German foodstuff it is, however, crucial for the underlying crop rotation and farming processes and was therefore not excludable from assessment.

RESPONSE SHEET– REVIEWER #1

Reitmayr et al. (1995) (as quoted in Stolze et al. 2000 , p. 55)			Germany	winter wheat	1.001	429	63%
				potatoes	1.153	958	
Tuomisto et al. (2012) SI table S.1	LCA modelling (data from previous studies)	Indirect ⁹ and direct inputs until farmgate; comparable with cradle to farmgate	UK	winter wheat	1.772	629	36%
Animal-based							84%
Basset-Mens, Werf (2005)	LCA modelling (data from other studies)	direct and indirect inputs and effects; comparable with cradle to farmgate	France (Bretagne)	pig	4236	4022	95%
Casey, Holden (2006)	LCA modelling (data from questionnaires and other studies)	cradle to farmgate	Ireland	beef	5346	2302	82%
Flessa et al. (2002)	modelling (on basis of empirical data and other studies)	direct inputs and limited ¹⁰ indirect inputs until farmgate; comparable with cradle to farmgate	Germany, Oberbayern (South)	beef/cattle	4177	3037	73%
Dairy							63%
Bos et al. (2007)	modelling (model DairyWise)	indirect and direct emissions; comparable with cradle to farmgate	Netherlands	dairy	11		61%
Dalgaard et al. (2006)	LCA modelling (based on empirical data from 2138 private farm accounts)	cradle to farmgate ¹²	Denmark	dairy; sandy soil	6.335	5.459	57%
				dairy; sandy loam soil	5.803	1.669	
Haas et al. (2001)	LCA modelling (based on empirical data from 35 farms in the region)	direct and upstream (indirect) processes; comparable with cradle to farmgate	Germany, Allgäu (Southern Bavaria)	dairy	9.400	6.300	67%
Thomassen et al. (2008)	LCI modelling (based on empirical data from field studies of 10 conventional and 11 organic farms, and data of previous studies)	cradle to farmgate	Netherlands	dairy	20.598	13.405	65%

⁹ Tuomisto et al. (2012) and Flessa et al. (2002) explicitly state that the production of farm buildings is not considered. However, as far as it was comprehensible, all other studies have similarly not included assessment of housing production.

¹⁰ Production of fertilizer was considered; other indirect inputs for precursors like pesticides and seeds were not included as they were considered negligible; infrastructure (machines and buildings) was not included. This studies system boundaries are least in line with our assessment scope but are still comparable due to the explanation as to why certain processes were excluded.

¹¹ As Bos et al. (2007) reports „GHG emissions per ha on the conventional dairy farms are 65% higher than on the organic model farms.” (p.3). We set organic as 100% and conventional as 165%.

¹² Authors refer to cradle to grave approach when introducing to the topic of LCA. They continue although with cradle to farmgate assessments of nitrogen surpluses, for example. The input data does also not include processes after farmgate. Therefore, we find this approach to be comparable with cradle to farmgate.

2. “It is also not clear, how authors estimated the emissions from organic farming based on these studies, and what was the level of uncertainty in the estimated values. I would like to see this description very clearly because it is crucial information on which the study is founded.”

Response: Thank you for pointing this out. We understand that this is an important part, which we did not elaborate clearly enough in the original version. When evaluating the regarding studies the following limitations arose:

Due to the varying estimation methods of considered studies, only four out of twelve papers report measures of deviation (Aguilera et al. 2015a and b, Thomassen et al. 2008, Basset-Mens, et al. 2005, Basset-Mens et al. 2005); Two studies give ranges for their found emission values (Tuomisto et al. 2012, Reitmayr et al. 1995); Four papers mention ranges or deviation for some input data but not their finally retrieved emission value (Casey et al. 2006, Flessa et al. 2002, Dalgaard et al. 2006, Haas et al. 2001); And lastly, three results are solely stated as definite values with no information about the uncertainty of these values (Cooper et al. 2011, Küstermann et al. 2008, Bos et al. 2007).

Because of this methodological inconsistency throughout the studies we found an adequate inclusion of statistical means like deviations not possible and therefore used another method of weighting the studies’ results for the calculation of an average.

The description of this approach is described in more detail now. The adjusted section now reads as follows:

Subsection 6.2.1, page 26 f.:

“As the selected studies are based on geophysical measurements and not on inferential statistics, a weighting based on the standard error of the primary study results like in standard meta-analysis⁸¹ was not possible. We aimed for a system that weights the underlying studies regarding their quality and therefore including their results weighted accordingly in our calculations. Within the scope of classic meta-analyses⁸² the studies’ individual quality is estimated according to their reported standard error (SE), which is understood as a measure of uncertainty: the smaller the SE, the higher the weight that is assigned to the regarding source. As, due to the varying estimation methods of considered studies, a majority of considered papers does not report measures of deviation for their results. These state definite values; therefore, there is no information about the precision of the results at hand. Against this background we have decided to use a modified approach to estimate the considered papers’ qualities⁸³. Following van Ewijk et al.⁸⁴ and Haase et al.⁸⁵ we apply three relevant context sensitive variables to approximate the standard error of the dependent variable and thereby evaluate the quality of each publication: The newer the paper (compared to the timeframe between 1968 and 2018) the higher we assume the quality of reported results. The more often

a paper was cited per year (measured on the basis of Google Scholar) the higher the paper's reputation. The higher the publishing journal's impact factor (measured with the SciMago journal ranking) the higher its reputation and therefore the paper's quality. For every paper the three indicators publishing year (shortened with PY in Table 2), citations/year (CY), and journal rank (SJR) rank a paper's impact on a scale from 1 to 10, where 1 describes the lowest qualitative rank and 10 the highest. The sum of these three factors (SUM) then determines the weight of a paper's result in the mean value (WEIGHT). The papers' reported emission-differences between organic and conventional (diff. org/conv) are weighted with the papers' specifically calculated WEIGHTS and finally aggregated to the emission difference between both systems.

With this approach we weight results of qualitatively valuable papers higher and are therefore able to reduce the level of uncertainty in the estimated values because standard errors could – due to inconsistencies in the underlying studies – not be used.”

3. “Authors have cited several of their previous works on which the current study is built upon. Unfortunately, it was very difficult to assess those studies. Particularly the reference 16 is only an abstract accepted for a conference. It makes it very challenging to know how the current approach is built upon the author's previous work and compare them.”

Response: We understand the reviewer's critique regarding this point. To respond to this comment, we have placed one of the references in another, newly formulated section. Another reference has been deleted. In detail, we proceeded as follows:

In the original version, we had referred to our previous work four times:

- *[former Reference 13] Gaugler, T., Rathgeber, A. & Stöckl, S. (2017)*
- *[former Reference 14] Gaugler, T. & Michalke, A. (2017)*
- *[former Reference 15] Michalke, A., Fitzer, F., Pieper, M., Kohlschütter, N. & Gaugler, T. (2019)*
- *[former Reference 16] Michalke, A., Gaugler, T. & Pieper, M. (2019)*

Based on the reviewer's comments the following adjustments have been made:

- *Due to the now more precisely explained underlying weighting method of the papers (please see our response to previous comment for detailed explanation) it is now more sensible to cite article “Gaugler, T., Rathgeber, A. & Stöckl, S. (2017)” [former Reference 13] there.*

- The article “Gaugler, T. & Michalke, A. (2017)” [former Reference 14] is a short piece [2 pages] that only focuses the environmental costs of nitrogen. We have used the following three years to comprehensively refine the methodology.
- The contribution “Michalke, A., Fitzer, F., Pieper, M., Kohlschütter, N. & Gaugler, T. (2019)” [former Reference 15] is a four-page conference paper submitted in November 2018 and presented as an oral presentation in March 2019. Based on the constructive suggestions we received at the conference, we have further developed the contribution to the state of this paper.
- “Michalke, A., Gaugler, T. & Pieper, M. (2019)” [former Reference 16] is another short conference contribution. While preparing this paper we assumed that the abstract would have been put online by the time of submission, but this has not been the case so far. Against this background we have now removed this reference.

4. “It is not clear whether authors include emissions associated with the transport of inputs such as fertilizers.”

Response: This indeed is an aspect, that should be more clearly addressed. In the original manuscript we wrote that “all resource inputs and outputs during production up to the point of selling by the primary producer are considered.” This does also include all emissions of transport along the whole value chain. Upstream supply chains (e.g. the production of fertilizer) and the emissions thereof are consistently viewed as part of the value chain. In the handbook for GEMIS (Fritsche, Schmidt. Handbuch zu GEMIS. 2008. http://www.iinas.org/tl_files/iinas/downloads/GEMIS/2008_g45_handbuch.pdf. p. 134) it says about the (in this case activated) ‘global switch’ of considering all transport processes: “If this switch is set, then all transports (also in upstream chains) are considered in the emission calculation of GEMIS. The global switch ‘non-stationary transport’ now determines whether stationary (non-stationary) transport processes are included or not. Non-stationary transport processes are ship, truck, train transports etc., while stationary transport processes are e.g. power lines and pipelines” [own translation from German to English].

Additional to transport emissions, we also consider emissions linked to the preliminary building of relevant infrastructure to be part of production inputs as we quantify the categories’ emission values.

Furthermore, we enhanced our calculations with including emissions from land use change (LUC) in the considered data. These emissions were not considered in GEMIS data. We therefore calculated those according to the frequently used method of Ponsion and Blonk (2012).

To meet the objection of the reviewer we have now removed previous ambiguity. These additional clarifications of the system boundaries are now added in the manuscript (new text passages are underlined):

Subsection 6.2.1, p. 22 ff.:

“This means that we consider all resource inputs and outputs during production up to the point of selling by the primary producer (“farmgate”). This includes emissions from all production-relevant transports as well as emissions linked to the preliminary building of production-relevant infrastructure. We specify that for animal products, emissions from feed production, as a necessary resource input, are assigned to these animal products. Such emissions naturally should include LUC emissions. LUC emissions are of negligible proportion for locally grown products, as agricultural land area is slightly decreasing in Germany³⁹. Thus, we have to focus solely on imported products; on imported feed for conventional animal and dairy products to be precise. Organic feed is not considered as article 14d of the EU-Eco regulation stipulates that organic farms have to primarily use feed which they produce themselves or which was produced from other organic farms in the same region⁷⁶. Even stricter rules are set by many of the organic farming associations in Germany, such as Bioland, Naturland or Neuland, that ban soymeal from Latin America completely⁷⁷. We assume that the emissions that could possibly be caused by organic farming in Germany through the import of feed constitute a negligibly small fraction of the total emissions of a product. Thus, no LUC emissions are calculated for organic products. For conventional products we calculate LUC emissions by application of the method of Ponsioen and Blonk³⁸. This method allows the calculation of LUC emissions for a specific crop in a specific country for a specific year. With regards to the year, we apply our reference year 2016. With regards to crop and country one has to keep in mind that in the case of Germany, the net imports of feed are the highest for soymeal, followed by maize and rapeseed meal, making up over 90% of all net positive feed imports⁷⁸. Maize and rapeseed meal are both imported mainly from Russia and Ukraine (93% and 87% of all imports⁷⁹). Taken together, the crop area of Russia and Ukraine is decreasing by 150,000 ha/year (data from 1990-2015 was used⁸⁰. Following Ponsioen and Blonk³⁸, we thus assume that there are no LUC emissions of agricultural products from these countries. This leaves us with soymeal, of which 97% are imported from Argentina and Brazil. We thus calculate LUC emissions of soymeal for Argentina and Brazil respectively. Data is used from Ponsioen and Blonk³⁸, except for data of the crop area, where updated data from FAOSTAT is used in order to match the reference year. We then weigh those country-specific emission values according to their import quantity. This results in 2.54 kgCO₂eq/kgSoymeal. To incorporate this value into the conventional emission data from GEMIS, we map the LUC emissions onto all the soymeal inputs connected to the food-specific products.”

5. “As authors already mention that yield in organic production is lower than the conventional, it may be argued that more land area will need to be cultivated for the same amount of production. However, this study does not cover the emissions resulting from the land-use conversion that will be required for this. I understand that incorporation of land-use change may complicate the study. At the same time, even if we assume that no LUC occurs in Germany, I may argue that the import of food will increase. Given that emissions that occur outside Germany are excluded in the study, it might be underestimating the emissions.”

Response: This is a very good argument that needs to be addressed thoroughly. In the original manuscript we mentioned the alleged effect of increasing imports that might result from a shift from conventional to organic farmland. However, we did not go into detail there. To correct for this informational insufficiency, we edited part of the discussion section to emphasize that an internalization of the external costs of each food category would prevent rising emissions potentially resulting from a widespread application of organic agriculture. That is because the prices of the most resource intensive products (which are animal-based foodstuff from both conventional and organic farming) would significantly increase and thereby lead to a reduced demand of such products due to price elasticity of demand of so called “normal goods” (which also include the examined foodstuff). The associated extensive land area of these products thus would become available for organic agriculture. Furthermore, there is evidence that a shift from conventional to organic practices would indeed be beneficial for the ecosystem services and long-term efficiency provided by the particular land area (Reganold et al. (1987), Reganold and Wachter (2016)).

A scenario-analysis is, however, not part of our study, in which we solely aim at examining the status quo. We therefore do not include a hypothetical emission increase from a shift to organic farming in our calculations.

Thank you, once again, for pointing out that this aspect of our paper needed more clarification. The adjusted section now reads as follows (new text passages are underlined):

Section 4, p. 16:

“Further doubt towards a transition to organic farming was spread by Smith et al.⁵² who rightfully addressed the potential increase of emissions resulting from a complete transition from conventional towards organic farming, given consumption patterns stay the same. These increases are thought to result from a higher amount of imported food, due to lower (regional) yields from organic farming. The financial incentives of internalization presented in our paper and the associated changing consumption patterns, however, pose a solution to these identified problems. Due to price elasticities of demand for food products (which are consistently regarded as ‘normal goods’ in economic literature), appropriate pricing of food would make products of organic production more competitive compared to their conventional

counterparts ⁵³: customers would increasingly opt for organic foodstuff due to the lowered price-gap between the two options. This could potentially press the boundaries of land use for agriculture as organic practices mostly require more land than conventional systems due to lower yields ⁵⁴⁻⁵⁶. However, our results suggest an increase in the prices of animal-based products to a significantly larger extent than the prices of plant-based products. The presumed consequential decline of animal-based product consumption would free an enormous landmass currently used for feed-production. Further expansion of area-intensive organic agriculture would subsequently be made possible ⁵⁷. Furthermore, there is evidence that a shift from conventional to organic practices would indeed be beneficial for the ecosystem services and long-term efficiency provided by the particular land area ^{7,58}.”

Due to the reviewer commenting on the significance of emission from land use conversion we have now enhanced our calculations with data from LUC (compare response to reviewer’s comment 4). However, these calculations still do not account for possible developments in the future. Our study focuses solely on the status quo of today’s agricultural conditions.

6. „The authors do not mention whether they take into account the temporal changes in emissions that can occur as the organic farming is followed on a continuous basis. It is very likely that the yield gaps between the conventional and organic farming may increase or decrease, and the same can happen with the soil-borne GHG emissions and soil carbon sequestration rates. Ideally, authors should account for uncertainties arising due to these, but if it is not possible due to the limitation of the data, these effects need to be discussed as a part of the uncertainty.“

Response: *Thank you for the reasonable suggestion that we need to clarify this point in the original version. As already stated in the answer to the previous commentary (comment 5), we solely aim at examining the status quo of German agricultural practices and have not investigated the temporal effects. We now point to this fact in our paper (Please refer both to the response to comment 5 for details and to (the new) subsection 6.4 entitled “Dealing with uncertainties”.) Following the suggestion of the reviewer, we have also added this passage (p. 19): “If one takes into account the temporal change in yield difference which would result by converting farms from conventional to organic farming, there is scientific consensus that the yield gap will decrease over time (Schrama et al. 2018, Sander and Hess 2019). Comparative studies between different cultivation methods also show that organic farming has lower soil-borne GHG emissions and higher rates of carbon sequestration in the soil (Scialabba and Müller-Lindenlauf 2010, Muller et al. 2017). Soil degradation resulting from conventional systems would slow down or could even be reversed by changing to organic farming (Küstermann et al. 2008, Azadi et al. 2011).”*

In order to further address possible uncertainties, we have decided to follow the reviewer's suggestion and also included a subsection for the discussion of uncertainties and assumptions that have been made throughout the study. Please find this as follows:

Subsection 6.4, p. 32 f.:

“6.4 Dealing with Uncertainties

Due to the interdisciplinarity and novelty of our study we connect several methodological approaches and refer to various sources for data. Against this background we had to accept some uncertainties while assembling and using the developed framework for our calculation. The studies included in our meta-analytical approach of calculating the difference between organic and conventional emission values, for one, are not fully consistent in the methodologies each of them uses (refer to SI.1 for details). Furthermore, from the results of all included studies it is apparent that there exists a wide range of emission differences between the farming practices, depending on the papers scope and examined produce⁹¹. We attempted to account for this through weighting the studies according to their fit regarding the object of research (compare subsection 6.2.1). Due to insufficient availability of data for the emission differences between organic and conventional on the basis of each narrow category an average for the emission difference was used. This possibly results in imprecisions during the internalization of the external costs on the level of all narrow categories. Therefore, we focus on the aggregated broad categories as this uncertainty can be evaded here. Furthermore, the in literature reported price factor for CO₂-equivalents is volatile over time impacting the results of this paper. It is to be expected that the external costs of GHG emissions are likely to rise in the future (compare section 2). Also, our study's scope is confined to the assessment of the current production situation within the German agricultural sector. Therefore, we do not account for future developments regarding a changing agricultural production landscape after internalization of the accounted external costs. We do, however, discuss possible effects on demand patterns as well as the environmental and social performance of the agricultural sector in section 4. Regarding the incorporated LUC emissions, there appears to be a lacking scientific consensus on a general method of calculation for such emissions^{38,92–94}. We thus want to emphasize that these additional emissions should be treated with caution and are thereby displayed separately from the other data.”

II. Format and language

1. “The format is not according to the journal, (methods are before the results).”

Response: Thank you for this reminder. The methods have now been moved to the end and can be found after the discussion/conclusion chapter. We have sensibly changed all necessary textual references according to the new format.

2. “Authors use words like ‘external effects’ and external costs, both of which I understand mean externality? I will suggest using the same term throughout to make it easy for the reader.”

Response: Thank you very much for this advice. Indeed, we used various terms synonymously until now making it rather difficult to understand for the reader. Therefore, we now use the terms in the following nuanced manner: When we talk about volume-related externalities measured in CO₂-equivalents, we now consistently use the term “externalities”. If it is talked about the follow-up costs measured in monetary units resulting from CO₂-eq. emission, we now refer to them as “external costs”.

An exception to this is the conscious use of the term “external climate costs” in the title as well as in the abstract of the paper. The addition of the word “climate” seems appropriate to us in these two cases as the reader should be informed briefly and concisely at first glance that the article addresses climate follow-up costs.

Furthermore, as we were quality checking our work with other colleagues in the field throughout this revision process it has come to our attention that the use of the term “production systems” when referring to either conventional or organic agricultural practices can lead to some confusion. We were advised that one might think about processing steps after the farmgate when reading this term. Therefore, we have changed all mentions in the text to “farming systems” to clarify referral to all steps before the farmgate.

3. “Page 3: what is chapter 3.2.1? I would prefer using ‘section’ instead of ‘chapter’.”

Response: We agree with the reviewer’s suggestion and now use the term ‘section’ when referring to a subchapter (e.g. ‘section 3.2.1’). We refer to a “section” as such only when referring to a chapter as a whole (e.g. section 5’).

4. “Page 3 paragraph 2, authors say that it will be technically incorrect to refer to the current approach as LCA and that the term carbon footprint is more appropriate. I, therefore, suggest authors talk about carbon footprint from the beginning.”

Response: We understand the reviewer’s critique relating to our use of the terms “Life Cycle Assessment” (LCA) and “carbon footprint” in chapter 2 (page 3). Indeed, our previous wording was not precise enough. LCA is a general approach for determining environmental impacts. This methodology can be used to determine the amount of various resources and pollutants arising during the production of a good (e.g. water consumption, SO₂ and CO₂

emissions, etc.) as well as their impact (e.g. for climate or human health). As our study focuses on the climate impact of agricultural products, which are quantified using CO₂-equivalents, we apply the methods of LCA on the food-specific emission quantities of CO₂-equivalents.

The term “carbon footprint” is merely a measure of the amount of CO₂-equivalents emitted. In short, by using Life Cycle Assessment it is possible to determine the carbon footprint of a product, in our case the various established food categories. Against this background, we propose that the description of LCA remains in the paper. Instead, we suggest that the expression “carbon footprint” (as the term used to describe the quantities of CO₂-equivalent emissions) should only be briefly discussed now. According to this more precise narrative we have now adapted the text as follows (new passages are underlined):

Subsection 6.1, p. 19:

“The quantification includes the determination of food specific GHG emissions – also known as carbon footprints³³ – occurring from cradle to farmgate by usage of a material flow analysis tool. Carbon footprints are understood within this paper in line with Pandey et al.⁶³ where all climate relevant gases, which in addition to CO₂ include, methane (CH₄), and nitrous oxide (N₂O), are considered. Their 100-year CO₂-equivalents conversion factors are henceforth defined as 28 and 265, respectively⁶⁹.”

5. „page 8: Authors mention that they use 11 food categories but name only 10.“

Response: Thank you for your careful reading. We accidentally forgot to list ‘cereals’ on the plant-based side of the eleven narrow food categories. This has been corrected and can be seen on page 9, subsection 3.1.

6. A) “Table 1, 2 and 3 need reformatting. Units are at the bottom instead of being at the top.”

Response: Thank you again for this formatting advice. We have now reformatted all tables according to comparable publication models in the journal. The new formatting can be seen below.

Subsection 3.1, p. 7:

Emission data (in kg CO ₂ eq/kgProduct)											
Broad categories [b]	prod. method			Narrow categories [n]	prod. method			Food-specific [i]	prod. method		
	conv. [E _{b,conv}]	with LUC	org. [E _{b,org}]		conv. [e _{b,n,conv}]	with LUC	org. [e _{b,n,conv}]		conv. [g _{b,n,i,conv}]	with LUC	org. [g _{b,n,i,org}]

RESPONSE SHEET– REVIEWER #1

Plant-based	0.20 /	0.11	Vegetables	0.04 /	0.02	field vegetables	0.03 /	0.02
						tomatoes	0.39 /	0.23
			Fruit	0.25 /	0.15	fruit	0.25 /	0.15
						rye	0.22 /	0.13
			Cereal	0.36 /	0.21	wheat	0.38 /	0.22
						oat	0.36 /	0.21
						barley	0.33 /	0.19
			Root Crops	0.06 /	0.04	potatoes	0.06 /	0.04
			Legumes	0.03 /	0.02	beans	0.03 /	0.02
			Oilseed	1.02 /	0.59	rapeseed	1.02 /	0.59
Animal-based	8.91 (13.39)	13.34	Eggs	1.24 (1.25)	1.86	eggs	1.24 (1.25)	1.86
			Poultry	13.16 (15.81)	19.71	broilers	13.16 (15.81)	19.71
			Ruminants	24.84 (36.95)	37.21	beef	24.84 (36.95)	37.21
			Pork	5.54 (9.56)	8.30	pork	5.54 (9.56)	8.30
Dairy	1.14 (1.59)	1.10	Milk	1.14 (1.59)	1.10	milk	1.14 (1.59)	1.10

Subsection 3.1, p. 8 f.:

Name	Country	Produce	D _{org/conv}	relevance				
				PY	CY	SJR	SUM	WEIGHT
Plant-based								
Aguilera et al. (2015)	Spain	citrus, fruits	49%	10	3	10	23	23%
Aguilera et al. (2015)	Spain	cereals, legumes, veg.	45%	10	3	10	23	23%
Cooper et al. (2011)	UK	crops	42%	9	2	2	13	13%
Küstermann et al. (2008)	Germany	arable	72%	8	3	4	15	15%
Reitmayr (1995)	Germany	wheat, potatoe	63%	6	1	1	8	8%
Tuomisto (2012)	EU	arable	36%	9	2	5	16	16%
			50%				98	100%
			58%					
Animal-based								
Basset-Mens; Werft (2005)	France	pig	95%	8	7	6	21	35%
Casey; Holden (2006)	Ireland	beef	82%	8	3	10	21	35%
Flessa et al. (2002)	Germany	beef/cattle	73%	7	5	6	18	30%
			84%				60	100%
			150%					
Dairy								
Bos et al. (2014)	Netherlands	dairy	61%	9	3	4	10	21%
Dalgaard et al. (2006)	Denmark	dairy	57%	8	2	6	16	21%
Haas et al. (2001)	Germany	dairy	67%	7	8	5	20	32%

RESPONSE SHEET– REVIEWER #1

Thomassen et al. (2008) Netherlands dairy

65%	8	10	6	24	26%
63%				70	100%
96%					

↓

Subsection 3.2, p. 11:

Broad categories [b]	prod. method						Narrow categories [n]	prod. method					
	Conv.			Org.				Conv.			Org.		
	pp _{b,conv} (€/kg Prod)	C _{b,conv} (€/kg Prod)	with LUC Δ _{b,conv}	pp _{b,conv} (€/kg Prod)	C _{b,conv} (€/kg Prod)	with LUC Δ _{b,conv}		pp _{b,n,conv} (€/kg Prod)	C _{b,n,conv} (€/kg Prod)	with LUC Δ _{b,n,conv}	pp _{b,n,org} (€/kg Prod)	C _{b,n,org} (€/kg Prod)	with LUC Δ _{b,n,org}
Plant-based	0.14	0.04	25%	0.39	0.02	5%	Vegetables	0.69	0.01	1%	1.10	~0.00	~0%
							Fruit	0.50	0.05	9%	0.57	0.03	5%
							Cereal	0.09	0.07	72%	0.31	0.04	12%
							Root Crops	0.08	0.01	14%	0.35	0.01	2%
							Legumes	0.02	0.01	33%	0.13	0.00	3%
							Oilseed	0.37	0.18	50%	0.42	0.11	25%
Animal-based	1.66	1.60 (2,41)	97% (146%)	3.41	2.40	70%	Eggs	1.21	0.22 (0,23)	18% (19%)	3.42	0.33	10%
							Poultry	1.72	2.37 (2,85)	137% (165%)	2.31	3.55	153%
							Ruminants	3.38	4.47 (6,65)	132% (197%)	3.90	6.70	172%
							Pork	1.35	1.00 (1,72)	74% (128%)	3.61	1.49	41%
Milk	0.26	0.21 (0,29)	78% (108%)	0.48	0.20	41%	Milk	0.26	0.21 (0,29)	78% (108%)	0.48	0.20	41%

6. B) “What is meant by field vegetables in table 1?”

Response: In GEMIS ‘field vegetables’ constitutes a collective term describing vegetables that are grown in the open air. This form of cultivation is in contrast to the horticultural cultivation of vegetables which uses greenhouses, foil tunnels or other artificially protected areas. We have added a footnote to the corresponding table explaining exactly this circumstance to clarify arising questions for the reader (Please refer to SI, former table 2, also to be found with comment 1).

Reviewer #2

1. “I suggest the Authors to rewrite the abstract with a clear presentation of the main motivations, objectives, methodologies and main insights (avoiding in the abstract terms as mark-up, maybe not a familiar term for the general readers of these topics).

Response: Thank you for this attentive comment. The abstract of course is crucial for the success of an article and therefore we have put careful effort into editing it according to your advice. We have now structured it along your suggested four subheadings (main motivations, objectives, methodologies, main insights) and have exchanged the unfamiliar term ‘mark-up’ with the more generally known and applicable ‘surcharge’. For faster reference we include the edited abstract in the following:

Abstract, p. 1:

“Although the agricultural sector is globally a main emitter of greenhouse gases, thorough economic analysis of environmental and social externalities has not yet been conducted. Available research especially lacks differentiation between farming systems and various food categories. A method addressing this scientific gap is established in this paper and applied in the context of Germany. Using LCA and meta-analytical approaches, we calculate the external climate costs of foodstuff. Results show that external greenhouse gas costs are highest for conventional animal-based products (2.41€/kg product; 146% surcharge on producer price level), followed by conventional dairy products (0.29€/kg product; 108% surcharge) and lowest for organic plant-based products (0.02€/kg product; 5% surcharge). The large difference of relative external climate costs between food categories as well as the absolute external climate costs of the agricultural sector imply the urgency for policy measures that close the gap between current market prices and the true costs of food.”

Furthermore, as we were quality checking our work with other colleagues in the field throughout this revision process it has come to our attention that the use of the term “production systems” - when referring to either conventional or organic agricultural practices - can lead to some confusion. We were advised that one might think about processing steps after the farmgate when reading this term. Therefore, we have changed all mentions in the text to “farming systems” to clarify referral to all steps before the farmgate.

2. “The literature review is weak. I suggest improve and update significantly this part.”

Response: Thank you a lot for this clear suggestion. As externality assessment in the agricultural context is a controversial topic and scientifically approached from various angles, it is indeed important to provide a profound overview of available sources and methodologies. We have enhanced the literature review following your note. Please find the edited version here with all new inputs underlined:

Section 2, p. 3 ff.:

“2. Research aim and literature review

[...]

There has been some scientific engagement previously, as Pretty et al. ¹⁴ set the scene for agricultural externality analysis at this century’s beginning: they were able to record significant environmental impacts of agriculture at the overall societal level in monetary terms for the United Kingdom. This approach was translated for other regions subsequently, with calculations of agricultural external costs for the United States and Germany ^{8,15}. However, these first external cost assessments, with their characteristic top-down approaches, did not link specific causal emission values with said costs. Yet, a bottom-up approach for monetizing externalities of country-specific agricultural reactive nitrogen emissions was later developed ¹⁶ and subsequently used for an external cost assessment of Dutch pig production ¹⁷. Despite, assessments concerning important agricultural emissions, which comprehensively differentiate between a variety of food categories, are yet missing. There exists a range of studies that quantify food-category-specific GHG emissions ¹⁸⁻²¹ while other studies disclose the difference of climate effects from conventional and organic practices (see table 2 for references). Monetizing such emissions, however, has been done for constituent food categories only ²². An encompassing connection between the quantification and monetization of GHG emissions differentiated by food categories and farming systems is what seems to be lacking in the currently available literature.

Congruent to methodological differences for monetizing agricultural greenhouse gases, there are also differences in the estimation level of greenhouse gas costs, which especially in the past have been vast. Prices per tonne of emission at the stock market, for example, are as low as 3.92 to 8.33 € during this study’s reference year ²³, whereas the IPCC in their last report of 2019 suggest a price between 135 and 5,500 \$ per tonne of CO₂-equivalents ²⁴. The German Federal Environmental Agencies (UBA) suggestion for the damage costs of GHG emissions also rose within the last years: in 2010 they suggested a rate of 80 € per tonne of CO₂-

equivalents²⁵, whereas this increased to 180 € per tonne in 2019²⁶. These great differences can be explained with methodological inconsistencies or a difference in approach, for example due to consideration of either damage or abatement costs. Furthermore, the price is expected to rise in the future²⁷, for example, describes that it must be “exceeding \$400 per tonne by mid-century” (p. 1271).

[...]

LCA has developed as a commonly used tool for examining material and substance flows of diverse products. Its origins lie in the analysis of energy flows but it is now commonly used to assess various processes²⁹. [...]

[...] Especially during the production of animal-based foodstuff livestock related gases like methane or nitrous oxide significantly contribute to the overall GHGs emitted⁴.

[...]”

Furthermore, we have clarified the use of the terms “carbon footprint” and “Life Cycle Assessment (LCA)” and put these in relation to the current scientific consensus. As our study focuses on the climate impact of agricultural products, which are quantified using CO₂-equivalents, we apply the methods of LCA on the food-specific emission quantities of CO₂-equivalents. The term “carbon footprint” is merely a measure of the amount of CO₂-equivalents emitted. In short, by using Life Cycle Assessment it is possible to determine the carbon footprint of a product, in our case the various established food categories. Against this background, we propose that the description of LCA remains in the paper. Instead, we suggest that the expression “carbon footprint” (as the term used to describe the quantities of CO₂-equivalent emissions) should only be briefly discussed now. According to this more precise narrative we have now adapted the text as follows (new passages are underlined):

Subsection 6.1, p. 19:

“The quantification includes the determination of food specific GHG emissions – also known as carbon footprints³² – occurring from cradle to farmgate by usage of a material flow analysis tool. Carbon footprints are understood within this paper in line with Pandey et al.⁶³ where all climate relevant gases, which in addition to CO₂ include, methane (CH₄), and nitrous oxide (N₂O), and their respective 100-year CO₂-equivalents conversion factors of 28 and 265, respectively, are considered.”

Besides these adaptations we have overall embedded the paper more strongly into current scientific literature throughout the whole text, which now refers to 94 - instead of the former 75 - references. Please find the answer to comment 7 for a more specific comparison between our findings and other works.

3. “In the current section 3 you need to support your options with other scientific papers already published. For example “...quantification and second the monetization of external effects from GHGs, visualized in...”. Why this approach and not other?. Other example, “...analysis tool GEMIS (Global Emission-model for Integrated Systems) 25 is used which offers...”. There are other approaches. Why not others? On other words, this section is too much descriptive in a part where you make several options.”

Response: We thank you for this very understandable commentary. Within the text we have now made several changes to explain in greater detail on which basis we designed the framework and why we decided on the methodological options that shape our work. As we have now rearranged the order of the manuscript according to the journal’s guidelines the corresponding text is now placed in section 6. Please find according additions underlined in the following:

Subsection 6.1, p. 18 ff.:

“6.1 Outline of the method

We differentiate between two steps within this method of calculating food-category specific externalities and the resulting external costs. These are first the quantification and second the monetization of externalities from GHGs (visualized in figure 3). We use this bottom-up approach following the example of Grinsven et al.¹⁶ who conducted a cost-benefit analysis of reactive nitrogen emissions from the agricultural sector. This two-stepped method also allows the adequately differentiated assessment for GHG emissions of various food categories.

The quantification includes the determination of food specific GHG emissions – also known as carbon footprints³² – occurring from cradle to farmgate by usage of a material flow analysis tool. Carbon footprints are understood within this paper in line with Pandey et al.⁶⁸ where all climate relevant gases, which (in addition to CO₂) include methane (CH₄) and nitrous oxide (N₂O), are considered. Their 100-year CO₂-equivalents conversion factors are henceforth defined as 28 and 265, respectively⁶⁹. Here, the material flow analysis tool GEMIS (Global Emission-model for Integrated Systems)³⁷ is used which offers data for a variety of conventionally farmed foodstuffs. As GEMIS data focuses on emissions from conventional agricultural systems we carried out the distinction to organic systems ourselves. We determined the difference in GHG emissions between the systems by applying meta-analytical methods to studies comparing the systems’ GHG emissions directly to one another. Meta-analysis is commonly used in the agricultural context, for example when comparing the productivity of both systems^{55–57} or their performance⁷.

For better communicability we first aggregate the 11 food specific datasets given in GEMIS to the broader food categories ‘plant-based’, ‘animal-based’, and ‘dairy’ by weighting them with their German production quantities (cf. section 3.1).

[...]

Figure 3: Visualization of the method of quantifying and monetizing product specific externalities; in the case of Germany, emission data was obtained from GEMIS³⁸, we used production data from the German Federal Statistical Office³⁷ and AMI^{42,64}, and calculated the emission difference between organic and conventional production based on a meta-analytical approach (see subsection 3.2.1); the category specific emission data was calculated on the basis of these input data; the emission cost rate was obtained from UBA²⁶; the category specific external costs were determined on the basis of the previously developed price-quantity-framework (see subsection 3.2.2). Source data are provided as a source data file.“

Furthermore, we now elaborate why we decided to use the tool GEMIS for our approach. Please find the corresponding passage as follows with all updates underlined:

Subsection 6.2.1, p. 21:

“Starting with the data on food-specific emissions, GEMIS is used because of its large database of life-cycle data on agricultural products with a geographic focus on Germany. GEMIS is a World-Bank acknowledged tool for their platform on climate-smart planning and draws on 671 references which are traced back to 13 different databases. The German Federal Environmental Agency uses GEMIS as a database for their projects and reports establishing it

to be an adequate tool for the German context especially ^{72,73}. This tool is provided by the International Institute for Sustainability Analysis and Strategy (IINAS). GEMIS offers a complete view on the life cycle of a product, from primary energy and resource extraction to the construction and usage of facilities and transport systems.”

The response to the next comment (4.) also entails more detail on our use of data and methodologies, concerning subsection 6.2. We have made an effort to link our approach to other scientific literature. Please find more information in the following response.

4. “On the other hand, the Authors need to be more specific in this section. For example, “...individual countries was accessible, EU-data is used.”. What does this mean?”

Response: Thank you very much for your comment. You are absolutely right, that subsection 6.2 (formerly 3.2) contains some formulations about the specifics of the data that might be confusing for readers. We clarified these passages. You can find a list of the adjusted passages below (new passages are underlined) along with a short explanation as to why we have specified the regarding passages:

The following explains about our use of GEMIS:

Section 6.2.1, p. 21:

“Starting with the data on food-specific emissions, GEMIS is used because of its large database of life-cycle data on agricultural products with a geographic focus on Germany. GEMIS is a World-Bank acknowledged tool for their platform on climate-smart planning and draws on 671 references which are traced back to 13 different databases. The German Federal Environmental Agency uses GEMIS as a database for their projects and reports establishing it to be an adequate tool for the German context especially ^{72,73}. This tool is provided by the International Institute for Sustainability Analysis and Strategy (IINAS). GEMIS offers a complete view on the life cycle of a product, from primary energy and resource extraction to the construction and usage of facilities and transport systems. In case GEMIS offered no data for Germany for certain foodstuff (this is the case for maize, milk and eggs), data from climatically comparable European countries is used. [...] In this study the system boundaries for assessing food-specific GHG emissions span from cradle to farmgate. This means that we consider all resource inputs and outputs during production up to the point of selling by the primary producer (“farmgate”). This includes emissions from all production-relevant transports as well as emissions linked to the preliminary building of production-relevant infrastructure. [...]”

The following specifies the system’s boundaries in greater detail than before:

Section 6.2.1, page 21 f.:

“In this study the system boundaries for assessing food-specific GHG emissions span from cradle to farmgate. This means that we consider all resource inputs and outputs during production up to the point of selling by the primary producer (“farmgate”). This includes emissions from all production-relevant transports as well as emissions linked to the preliminary building of production-relevant infrastructure. We specify that for animal-based products, emissions from feed production, as a necessary resource input, are assigned to these animal-based products.”

As explained in response to comment 5. we now include the emissions of LUC in our calculations. The reasoning and approach for this is explained in the following:

Section 6.2.1, page 22 f.:

“Such emissions naturally should include LUC emissions. LUC emissions are of negligible proportion for locally grown products, as agricultural land area is slightly decreasing in Germany³⁹. Thus, we have to focus solely on imported products; on imported feed for conventional animal-based and dairy products to be precise. Organic feed is not considered as article 14d of the EU-Eco regulation stipulates that organic farms have to primarily use feed which they produce themselves or which was produced from other organic farms in the same region⁷¹. Even stricter rules are set by many of the organic farming associations in Germany, such as Bioland, Naturland or Neuland, that ban soymeal from Latin America completely⁷². We assume that the emissions that could possibly be caused by organic farming in Germany through the import of feed constitute a negligibly small fraction of the total emissions of a product. Thus, no LUC emissions are calculated for organic products. For conventional products we calculate LUC emissions by application of the method of Ponsioen and Blonk³⁸. This method allows the calculation of LUC emissions for a specific crop in a specific country for a specific year. With regards to the year, we apply our reference year 2016. With regards to crop and country one has to keep in mind that in the case of Germany, the net imports of feed are the highest for soymeal, followed by maize and rapeseed meal, making up over 90% of all net positive feed imports⁷⁸. Maize and rapeseed meal are both imported mainly from Russia and Ukraine (93% and 87% of all imports⁷⁹). Taken together, the crop area of Russia and Ukraine is decreasing by 150,000 ha/year (data from 1990-2015 was used⁷⁵). Following Ponsioen and Blonk³⁸, we thus assume that there are no LUC emissions of agricultural products from these countries. This leaves us with soymeal, of which 97% are imported from Argentina and Brazil. We thus calculate LUC emissions of soymeal for Argentina and Brazil respectively. Data is used from Ponsioen and Blonk³⁸, except for data of the crop area, where

updated data from FAOSTAT is used in order to match the reference year. We then weigh those country-specific emission values according to their import quantity. This results in 2.54 kgCO₂eq/kgSoymeal. To incorporate this value into the conventional emission data from GEMIS, we map the LUC emissions onto all the soymeal inputs connected to the food-specific products.

[...]

For this weighting and aggregation step, only production quantities used for human nutrition were considered, thus feed and industry usage of food are ruled out (in contrast to emission calculation, where feed is indeed considered). Besides the German Federal Office of Statistics³⁶, source for this data is the German Society for Information on the Agricultural Market (AMI)^{41,70}. Only production data for conventional production is used. Thereby we imply equal ratios of production quantities across the food categories. This does not fully reflect the current situation of organic production properties but allows for a fair comparison between emission data of organic and conventional food categories. In table 3 all production data is listed, whereby total production quantities in 1,000t can be found in the right column. Translating these into percentage shares, the column right to the narrow category's column represents the shares of the specific foods inside the narrow categories, whereas the column right to the broad category's column represents the shares of the narrow categories inside the broad categories. These shares are expressed in formula 2a and 2b (sub-section 6.3.1) by the terms $\frac{p_{b,n,conv}}{P_{b,conv}}$ (share in broad categories) and $\frac{q_{b,n,i,conv}}{P_{b,n,conv}}$ (share in narrow categories).

[table 4]

We aggregate GEMIS emission data ($q_{b,n,i,conv}$) to narrow ($e_{b,n,conv}$) and broad categories ($E_{b,conv}$) by multiplying the respective emission data with the shares from table 3 (cf. formula 2a & b, subsection 6.3.1). From these conventional emission values we derive emissions for organic production. For narrow as well as broad categories, the respective conventional emission values are multiplied with the applicable emission-differences ' $D_{b, org/conv}$ ' (cf. table 2).“

The next section specifies the meta-analytical approach we used to calculate the differences between the systems of organic and conventional agriculture:

Section 6.2.1, p. 25 ff.:

“To cover a reasonably relevant period, we decided to search for studies published within the past 50 years (from 1969 to 2018) and could therefore identify fifteen relevant studies, spanning from 1995 to 2015. Four of these studies have Germany as their reference country while the other eleven focus on other European countries (Denmark, France, Ireland, Netherlands, Spain, UK; please consult SI.1 for specifics). The weighted mean of the individual study results amounts to the difference in GHG emissions between the two farming production systems. As the selected studies are based on geophysical measurements and not on inferential statistics, a weighting based on the standard error of the primary study results like in standard meta-analysis⁸¹ was not possible. We aimed for a system that weights the underlying studies regarding their quality and therefore including their results weighted accordingly in our calculations. Within the scope of classic meta-analyses⁸² the studies’ individual quality is estimated according to their reported standard error (SE), which is understood as a measure of uncertainty: the smaller the SE, the higher the weight that is assigned to the regarding source. As, due to the varying estimation methods of considered studies, a majority of considered papers does not report measures of deviation for their results. These state definite values; therefore, there is no information about the precision of the results at hand. Against this background we have decided to use a modified approach to estimate the considered papers’ qualities⁸³. Following van Ewijk et al.⁸⁴ and Haase et al.⁸⁵ we apply three relevant context sensitive variables to approximate the standard error of the dependent variable and thereby evaluate the quality of each publication: The newer the paper (compared to the timeframe between 1968 and 2018) the higher we assume the quality of reported results. The more often a paper was cited per year (measured on the basis of Google Scholar) the higher the paper’s reputation. The higher the publishing journal’s impact factor (measured with the SciMago journal ranking) the higher its reputation and therefore the paper’s quality. For every paper the three indicators publishing year (shortened with PY in Table 2), citations/year (CY), and journal rank (SJR) rank a paper’s impact on a scale from 1 to 10, where 1 describes the lowest qualitative rank and 10 the highest. The sum of these three factors (SUM) then determines the weight of a paper’s result in the mean value (WEIGHT). The papers’ reported emission-differences between organic and conventional (diff. org/conv) are weighted with the papers’ specifically calculated WEIGHTS and finally aggregated to the emission difference between both systems.

With this approach we weight results of qualitatively valuable papers higher and are therefore able to reduce the level of uncertainty in the estimated values because standard errors could –

due to inconsistencies in the underlying studies – not be used. The results of this meta-analytical approach are listed in table 2 (subsection 3.1), further details can be found in SI (SI.1). The studies considered compare GHG emissions of farming systems in relation to the crop/farm area. However, since our study aims to compare GHG emissions in relation to the weight of foodstuff, we include the difference in yield (“yield gap”) between the two farming systems for plant-based products, and the difference in productivity (“productivity gap”) for animal-based and dairy products. For plant-based products the yield gap is 117%, meaning that conventional farming produces 17% more plant-based products than organic farming on a given area. This gap was derived from three comprehensive meta studies^{54–56} and weighted as just described for the emission difference between organic and conventional farming. For animal-based as well as dairy products the productivity gap could be determined with the same studies used for the meta-analytical estimation of the emission-differences^{49–51,85–87}. The productivity gap is 179% for animal-based and 153% for dairy products. In line with Sanders and Hess⁶⁰ the yield (or productivity) difference $\frac{yield_{conv}}{yield_{org}}$ affects the calculation of the food-weight-specific emission difference $\frac{GHG_{org\ food\ weight}}{GHG_{conv\ food\ weight}} = D_{org/conv}$ between both farming systems: The yield difference is hereby multiplied with the cropland-specific emission difference $\frac{GHG_{org\ cropland}}{GHG_{conv\ cropland}}$. Resulting from this, the emission difference can be formulated as follows:

$$D_{org/conv} = \frac{GHG_{org\ food\ weight}}{GHG_{conv\ food\ weight}} = \frac{GHG_{org\ cropland}}{GHG_{conv\ cropland}} \times \frac{yield_{conv}}{yield_{org}}$$

If the yield difference would not be included, emissions from organic farming would appear lower than they actually are as organic farming has lower emissions per kg of foodstuff but also lower yields per area. With formula 1, we adjust for that.”

The following section explains our used monetary data in better detail:

Section 6.2.2, p. 28:

“For the pricing of the food categories, we determine the total amount of proceeds that farmers accumulate for their sold foodstuff in €⁷¹ for each category (“producer-price”) divided by its total production quantity. Thereby we calculate the relative price per ton for

each foodstuff. We solely refer to producer prices as the system boundaries only reach until the farmgate.”

5. “Yet in the section 3, sometimes, it is hard to understand what part of the data was obtained by you and what part was obtained from other sources. For example, only in the subsection 3.2.1. we understand that “...basis of quantity- and emission-trend was conducted in order to align the data with the...”. How you did this? And the scientific support?”

Response: We agree with the reviewer’s evaluation. In fact, it was unclear at several instances within the original text, which components originate from what other sources and which components are original to us. For this reason, we have now made clear throughout the whole chapter from where we gathered the individual parts.

Furthermore, we have enhanced the current methodology with including emissions from land use change due to the notes of the reviewers. This calculation is based on the method of Ponsioen and Blonk. The addition to our approach is also described in section 3.

In order to facilitate tracking of the changes made according to the underlying comment we present the original chapter 3 in the following with all pertinent changes underlined for the reviewer:

Section 6, p. 19 ff.:

“6. Method and Data

[...]

6.1 Outline of the method

We differentiate between two steps within this method of calculating food-category specific externalities and the resulting external costs. These are first the quantification and second the monetization of externalities from GHGs (visualized in figure 3). We use this bottom-up approach following the example of Grinsven et al. ¹⁶ who conducted a cost-benefit analysis of reactive nitrogen emissions from the agricultural sector. This two-stepped method, however, also allows the adequately differentiated assessment for GHG emissions of various food categories.

The quantification includes the determination of food specific GHG emissions – also known as carbon footprints ³² – occurring from cradle to farmgate by usage of a material flow analysis tool. Carbon footprints are understood within this paper in line with Pandey et al. ⁶⁸ where all climate relevant gases, which (in addition to CO₂) include methane (CH₄) and nitrous oxide (N₂O), are considered. Their 100-year CO₂-equivalents conversion factors are henceforth defined as 28 and 265, respectively ⁶⁹. Here, the material flow analysis tool

GEMIS (Global Emission-model for Integrated Systems) ³⁷ is used which offers data for a variety of conventionally farmed foodstuffs. As GEMIS data focuses on emissions from conventional agricultural systems we carried out the distinction to organic systems ourselves. We determined the difference in GHG emissions between the systems by applying meta-analytical methods to studies comparing the systems' GHG emissions directly to one another. Meta-analysis is commonly used in the agricultural context, for example when comparing the productivity of both systems ⁵⁴⁻⁵⁶ or their performance ⁷.

[...]

[Figure 3]

Figure 3: Visualization of the method of quantifying and monetizing product specific externalities; in the case of Germany, emission data was obtained from GEMIS ³⁸, we used production data from the German Federal Statistical Office ³⁷ and AMI ^{42,64}, and calculated the emission difference between organic and conventional production based on a meta-analytical approach (see subsection 3.2.1); the category specific emission data was calculated on the basis of these input data; the emission cost rate was obtained from UBA ²⁶; the category specific external costs were determined on the basis of the previously developed price-quantity-framework (see subsection 6.2.2). Source data are provided as a source data file.

6.2 Input data

In the following, we differentiate between input data for the quantification and for the monetization of external effects. The reference year for this analysis is 2016 and the reference country is Germany, which is listed as the third most affected country in the 'Global Climate Risk Index 2020' Ranking ⁶⁵.

6.2.1 Input data for quantification

Input data for quantification includes data on the food-specific amount of CO₂ emissions during the conventional farming process from the material flow analysis tool GEMIS ³⁸. For the meta-analytical methods, used to translate assessed emissions to organic systems, we gather data on the difference in emissions between conventional and organic farming production systems. [...] We specify that for animal products, emissions from feed production, as a necessary resource input, are assigned to these animal products. Such emissions naturally should include LUC emissions. LUC emissions are of negligible proportion for locally grown products, as agricultural land area is slightly decreasing in Germany ⁴⁰. Thus, we have to focus solely on imported products; on imported feed for conventional animal and dairy products to be precise. Organic feed is not considered as article 14d of the EU-Eco regulation stipulates that organic farms have to primarily use feed which they produce themselves or which was produced from other organic farms in the same region ⁷⁰. Even stricter rules are set by many of the organic farming associations in Germany, such as

Bioland, Naturland or Biopark, that ban soymeal from Latin America completely. We assume that the emissions that could possibly be caused by organic farming in Germany through the import of feed even despite the EU-Eco regulation constitute a negligibly small fraction of the total emissions of a product. Thus, no LUC emissions are calculated for organic products. For conventional products we calculate LUC emissions by application of the method from Ponsioen and Blonk³⁹. This method allows the calculation of LUC emissions for a specific crop in a specific country for a specific year. With regards to the year, we apply our reference year 2016. With regards to crop and country one has to keep in mind, that in the case of Germany, the net imports of feed are the highest for soymeal, followed by maize and rapeseed meal, making up over 90% of all net positive feed imports⁷¹. Maize and rapeseed meal are both imported mainly from Russia and Ukraine (93% and 87% of all imports⁷²). Taken together, the crop area of Russia and Ukraine is decreasing by 150,000 ha/year (data from 1990-2015 was used⁷³). Following the methodology by Ponsioen and Blonk³⁹, we thus assume that there are no LUC emissions of agricultural products from these countries. This leaves us with soymeal, of which 97% are imported from Argentina and Brazil. We thus calculate LUC emissions of soymeal for Argentina and Brazil respectively. Data is used from Ponsioen and Blonk³⁹, except for data of the crop area, where updated data from FAOSTAT is used in order to match the reference year. We then weigh those country-specific emission values according to their import quantity. This results in 2.54 kgCO₂eq/kgSoymeal. To incorporate this value into the conventional emission data from GEMIS, we map the LUC emissions onto all the soymeal inputs connected to the food-specific products.

For aggregation to narrow categories, we categorize every dataset from GEMIS into one of the eleven narrow food categories. The choice of separation into these specific categories is based on the categorization of the German Federal Office of Statistics³⁷ from which production data was obtained. According to one category's yearly production quantity, we incorporate every food product into the weighted mean of its corresponding food category. [...] Only production data for conventional production is used. Thereby we imply equal ratios of production quantities across the food categories. This does not fully reflect the current situation of organic production properties but allows for a fair comparison between emission data of organic and conventional food categories. In table 3 all production data is listed, whereby total production quantities in 1,000t can be found in the right column. Translating these into percentage shares, the column right to the narrow category's column represents the shares of the specific foods inside the narrow categories, whereas the column right to the broad category's column represents the shares of the narrow categories inside the broad

categories. These shares are expressed in formula 2a and 2b by the terms $\frac{p_{b,n,conv}}{P_{b,conv}}$ (share in broad categories) and $\frac{q_{b,n,i,conv}}{p_{b,n,conv}}$ (share in narrow categories).

[Table 4]

Table 4: Production data [$q_{b,n,i,conv}$] for food-specific products and share in broad and narrow categories for 2016 in Germany; production data was obtained from the German Federal Office of Statistics ³⁶ and AMI ^{41,70}. Source data are provided as a source data file.

We aggregate GEMIS emission data ($q_{b,n,i,conv}$) to narrow ($eb_{n,conv}$) and broad categories ($E_{b,conv}$) by multiplying the respective emission data with the shares from table 3 (cf. formula 2). From these conventional emission values we derive emissions for organic production. For narrow as well as broad categories, the respective conventional emission values are multiplied with the applicable emission-differences ‘ $Db_{org/conv}$ ’ (cf. table 2).

With this data we aggregate the above mentioned eleven food-categories to three broad categories: ‘plant-based’, ‘animal-based’ and ‘dairy’. [...]

As mentioned before, only data regarding externalities of conventional agricultural production is included in GEMIS and could therefore be aggregated. Nevertheless, by applying meta-analytical methods regarding the percentage difference of GHG emissions between conventional and organic production, we derive emission data for organic production for each of the broad categories (plant-based, animal, and dairy). It has to be noted that LUC emissions are consistently excluded from this procedure. To derive emission differences between organic and conventional farming, research was conducted by snowball sampling from already existing and thematically fitting meta-analysis, by keyword searching in research databases, as well as forward and backward search on the basis of already known sources. [...] As the selected studies are based on geophysical measurements and not on inferential statistics, a weighting based on the standard error of the primary study results like in standard meta-analysis ⁷⁴ was not possible. We aimed for a system that weights the underlying studies regarding their quality and therefore including their results weighted accordingly in our calculations. Within the scope of classic meta-analyses ⁷⁵ the studies’ individual quality is estimated according to their reported standard error (SE), which is understood as a measure of uncertainty: the smaller the SE, the higher the weight that is assigned to the regarding source. As, due to the varying estimation methods of considered studies, a majority of considered papers does not report measures of deviation for their results. These state definite values;

therefore, there is no information about the precision of the results at hand. Against this background we have decided to use a modified approach to estimate the considered papers' qualities ⁷⁶. Following van Ewijk et al. ⁷⁷ and Haase et al. ⁷⁸ we apply three relevant context sensitive variables to approximate the standard error of the dependent variable and thereby evaluate the quality of each publication: The newer the paper (compared to the timeframe between 1968 and 2018) the higher we assume the quality of reported results. The more the paper was cited per year (measured on the basis of Google Scholar) the higher the paper's reputation. The higher the publishing journal's impact factor (measured with the SciMago journal ranking) the higher its reputation and therefore the paper's quality. For every paper the three indicators publishing year (shortened with PY in Table 2), citations/year (CY), and journal rank (SJR) rank a paper's impact on a scale from 1 to 10, where 1 describes the lowest qualitative rank and 10 the highest. The sum of these three factors (SUM) then determines the weight of a paper's result in the mean value (WEIGHT). The papers' reported emission-differences between organic and conventional (diff. org/conv) are weighted with the papers' specifically calculated WEIGHTS and finally aggregated to the emission difference between both systems.

With this approach we weight results of qualitatively valuable papers higher and are therefore able to reduce the level of uncertainty in the estimated values because standard errors could – due to inconsistencies in the underlying studies – not be used. The results of this meta-analytical approach are listed in table 2 (subsection 3.1), further details can be found in SI. The studies considered compare GHG emissions of farming systems in relation to the crop/farm area. However, since our study aims to compare GHG emissions in relation to the weight of foodstuff, we include the difference in yield (“yield gap”) between the two farming systems for plant-based products, and the difference in productivity (“productivity gap”) for animal and dairy products. For plant-based products the yield gap is 117%, meaning that conventional farming produces 17% more plant-based products than organic farming on a given area. This gap was derived from three comprehensive meta studies ^{55–57} and weighted as just described for the emission difference between organic and conventional farming. For animal as well as dairy products the productivity gap could be determined with the same studies used for the meta-analytical estimation of the emission-differences ^{50–52,78–80}. The productivity gap is 179% for animal and 153% for dairy products. In line with Sanders and Hess ⁸¹ the yield (or productivity) difference $\frac{yield_{conv}}{yield_{org}}$ affects the calculation of the food-weight-specific emission difference $\frac{GHG_{org\ food\ weight}}{GHG_{conv\ food\ weight}} = D_{org/conv}$ between both farming

systems: The yield difference is hereby multiplied with the cropland-specific emission-difference $\frac{GHG_{org\ cropland}}{GHG_{conv\ cropland}}$. Resulting from this, the emission difference can be formulated as follows:

$$D_{org/conv} = \frac{GHG_{org\ food\ weight}}{GHG_{conv\ food\ weight}} = \frac{GHG_{org\ cropland}}{GHG_{conv\ cropland}} \times \frac{yield_{conv}}{yield_{org}}$$

If the yield difference would not be included, emissions from organic farming would appear lower than they actually are as organic farming has lower emissions per kg of foodstuff but also lower yields per area. With formula 1, we adjust for that.

6.2.2 Input data for monetization

[...] Following UBA, these damage costs are analyzed in the following categories: agriculture, forestry, sea level rise, cardiovascular and respiratory disorders related to cold and heat stress, malaria, dengue fever, schistosomiasis, diarrhea, energy consumption, water resources, and unmanaged ecosystems⁸². [...]

For the pricing of the food categories, we determine the total amount of proceeds that farmers accumulate for their sold foodstuff in €⁴¹ for each category (“producer-price”) divided by its total production quantity. Thereby we calculate the relative price per ton for each foodstuff. We solely refer to producer prices as the system boundaries only reach until the farmgate.

6.2.3 Calculating output data

[...] Combining the input data, we are now able to quantify and monetize externalities of GHGs for different food categories.

6.3.1 Calculating output data of quantification: category-specific GHG emissions

For quantification we separate between the following two steps: First, the aggregation of emissions data to broader categories and second the differentiation between conventional and organic farming systems. We iterate these steps two times, once for broad categories of animal products, plant-based products and dairy and once for more narrow categories of vegetables, fruits, root crops, legumes, cereal and oilseeds on the plant-based side as well as milk, eggs, poultry, ruminant and pig on the animal-based side. [...]

[figure 4]

Figure 4: Visualization of the quantification process and corresponding input and output data. GEMIS data ³⁷ ($g_{b,n,i,conv}$) and production data ^{36,41,70} ($q_{b,n,i,conv}$) are combined and emission data for broad ($E_{b,conv}$) and narrow ($e_{b,n,conv}$) categories is derived for conventional production. Organic emission values are calculated by multiplication of conventional emission values ($E_{b,org}$ and $eb_{n,org}$) with the emission difference ($D_{b,org/conv}$) (cf. subsection 6.2.1). Source data are provided as a source data file.

[...]"

In addition to the adjustments described above, we would like to discuss the last part of the reviewer’s commentary (“... in the subsection 3.2.1. we understand that “...basis of quantity- and emission-trend was conducted in order to align the data with the...”. How you did this? And the scientific support?”) in the following:

Since agricultural emission quantities are only available until 2015 and since they are subject to significant fluctuations over time, we have decided to extrapolate the historical data to the reference year of 2016. The following historical data form the basis of our calculation:

GHG-emissions [Mio tCO₂-eq] in Germany per year:

2000	67,6
2001	67,1
2002	65
2003	64,1
2004	64
2005	63,4
2006	62,6
2007	62
2008	64,3
2009	63,7
2010	62,9
2011	64,5
2012	64,1
2013	65,2
2014	66,6
2015	67

(Source: National trend tables for German reporting of atmospheric emissions 1990 - 2016 [Nationale Trendtabellen für die deutsche Berichterstattung atmosphärischer Emissionen

1990 - 2016]; <https://www.umweltbundesamt.de/daten/klima/treibhausgas-emissionen-in-deutschland#minderungsziele-fur-treibhausgase>)

These values were extrapolated to the year 2016 by means of linear regression, which in our opinion can be regarded as a standard procedure of statistics in order to relate the underlying data to the reference year. (cf. http://ci.columbia.edu/ci/premba_test/c0331/s7/s7_6.html) (Lane 2017, p. 462)

The resulting linear equation is determined as $y = -0,0099x + 64,715$, with which we can calculate the value for 2016 (set as 'x'), which is 64,4526 Mio tCO₂ -eq extrapolated. Thus, there is a slightly declining trend of emissions, whereby 2010 emissions (in the trend function) are 99,91% of those in 2016. GEMIS data from 2010 is accordingly multiplied with this percentage value.

Our suggestion is to include the data concerning linear interpolation in the SI – if the reviewer find this necessary – and thus enable the reader to understand the calculation in detail.

Following the reviewer’s advice, we added “Lane 2017, p. 462” as reference in the main text (subsection 6.2.1, p. 21.)

6. “In turn, sometimes, it is, also, hard understand the importance of each equation. For example, improve the explanations about the equation (1) and the relevance of this equation for your research. The same for the others. Another question is about the source of these equations that need to be clarified.”

Response: Thank you for this comment. It is very important to us that the equations enhance rather than hinder the understanding of the operations in this paper. Therefore, we improved the textual explanations of formula 1, 2, 3, 4 and 5 and, when necessary, we also added literature references (for formula 1). We also decided to remove formula 3, 4 and 5 as a sole textual explanation seemed to be sufficient in these cases. In the following, changes of the original text are underlined (for formulas 6-13 refer to reviewer’s comment 7):

Formula 1: We adjusted the description & added a literature reference:

Subsection 6.2.1, p. 27:

“In line with Sanders and Hess⁸² the yield (or productivity) difference $\frac{\square\square\square\square\square\square\square}{\square\square\square\square\square}$ affects the calculation of the food-weight-specific emission difference $\frac{\square\square\square\square\square\square\square\square\square\square}{\square\square\square\square\square\square\square\square\square}$ = $\square\square\square\square\square$ between both production systems: The yield difference is hereby multiplied with the cropland-specific emission-difference $\frac{\square\square\square\square\square\square\square\square\square}{\square\square\square\square\square\square\square\square\square}$. Resulting from this, the emission difference can be formulated as follows:

Formula 3: We erased this formula as the operation is easily understandable with sole textual explanation; we also added direct language to emphasize that these are our own calculations:

Subsection 6.3.1, p. 31:

“In the second step, we calculate emission values for organic production by multiplying the calculated emission-difference ‘ $D_{b,org/conv}$ ’ between both farming systems (cf. subsection 6.2.1) with the conventional emission values. These organic emission values are denoted as ‘ $E_{b,org}$ ’ for broad categories and ‘ $e_{b,n,org}$ ’ for narrow categories.”

Formula 4 & 5: We deleted these formulas, as textual description seems to be sufficient for understanding the calculations; we added direct language to emphasize that these are our own calculations; also, indices are adjusted:

Subsection 6.3.2, p. 31:

“To calculate the monetary cost ‘ C_b ’ of category-specific emissions, we multiply the cost rate ‘ P ’ for CO₂-equivalent emissions with the category-specific emission data ‘ E_b ’ or ‘ $e_{b,n}$ ’ (depending on whether broad or narrow categories are observed). Further, we determine percentage surcharge costs ‘ Δ_{\square} ’ by setting this cost in relation to the producer price ‘ pp_b ’ of the respective food-category: $\Delta_{\square} = \frac{\square_{\square}}{\square_{\square}}$ (the calculations are analogues for narrow categories). These surcharge costs represent the price increase necessary to internalize all externalities from GHG emissions for a specific food-category. “

7. “In this section 4 you need to link more the results obtained with the equations and methodologies presented before and to compare more specifically these results with other works.”

Response: Thank you for this comment. To ensure that the link between the methodology and the results is clear, we added the symbols and indices (which we introduced in the methods) in all the tables 1, 2, 3 and 4 as well as figure 1, 2 (formerly table 5, 6) and figure 4 (formerly figure 2). Furthermore, we edited the formula in section 3 (formerly section 4). All formula in section 3 (formerly known as formula 8, 9, 10, 11, 12 and 13) are now only described in the text and do not appear separately anymore. The reason behind this is that a sole textual description seemed to be sufficient for understanding in all cases, whereas the excessive listing of formulas might confuse the reader. By also improving the description of all these calculations, we further tried to link the results to the methods. In the order that they appear in the manuscript, these changes are chronologically listed in the following (changes are underlined):

Table 1: We changed the style of the table and added the according symbols and indices to the presented data; we also expanded the table description.

Subsection 3.1, p. 6 f.:

Emission data (in kg CO ₂ eq/kgProduct)											
Broad categories [b]	prod. method			Narrow categories [n]	prod. method			Food-specific [i]	prod. method		
	conv. [E _{b,conv}]	with LUC	org. [E _{b,org}]		conv. [e _{b,n,conv}]	with LUC	org. [e _{b,n,conv}]		conv. [g _{b,n,i,conv}]	with LUC	org. [g _{b,n,i,org}]
Plant-based	0.20 /		0.11	Vegetables	0.04 /		0.02	field vegetables	0.03 /		0.02
				tomatoes				0.39 /		0.23	
				Fruit	0.25 /		0.15	fruit	0.25 /		0.15
				rye				0.22 /		0.13	
				Cereal	0.36 /		0.21	wheat	0.38 /		0.22
				oat				0.36 /		0.21	
				barley				0.33 /		0.19	
				Root Crops	0.06 /		0.04	potatoes	0.06 /		0.04
Legumes	0.03 /		0.02	beans	0.03 /		0.02				
Oilseed	1.02 /		0.59	rapeseed	1.02 /		0.59				
Animal-based	8.91 (13.39)		13.34	Eggs	1.24 (1.25)		1.86	eggs	1.24 (1.25)		1.86
				Poultry	13.16 (15.81)		19.71	broilers	13.16 (15.81)		19.71
				Ruminants	24.84 (36.95)		37.21	beef	24.84 (36.95)		37.21
				Pork	5.54 (9.56)		8.30	pork	5.54 (9.56)		8.30
Dairy	1.14 (1.59)		1.10	Milk	1.14 (1.59)		1.10	milk	1.14 (1.59)		1.10

Table 1: Emission data for food-specific, narrow and broad categories; Emission data for food-specific, narrow and broad categories (following the classification from the German Federal Office of statistics ³⁶); food-specific emission data for conventional production was derived from Global Emissions Model for Integrated Systems (GEMIS) ³⁷ and aggregated to narrow and broad categories with German production data ³⁶; differentiation between conventional and organic production was derived with a meta-analytical approach (for details refer to the methods section and the Supplementary Information (SI), SI.1); LUC data is approximated to be the LUC emissions of soymeal fodder, emissions of it are calculated with the method of Ponsioen and Blonk ³⁸. Emission data including LUC emissions are shown in brackets. Source data are provided as a source data file.

Table 2: As with table 1, we changed the style of the table and added the according symbols and indices to the presented data; we also expanded the table description.

Subsection 3.1, p. 8 f.:

Name	Country	Produce	D _{org/conv}	relevance						
				PY	CY	SJR	SUM	WEIGHT		
Plant-based										
Aguilera et al. (2015)	Spain	citrus, fruits	49%	10	3	10	23	23%		
Aguilera et al. (2015)	Spain	cereals, legumes, veg.	45%	10	3	10	23	23%		
Cooper et al. (2011)	UK	crops	42%	9	2	2	13	13%		
Küstermann et al. (2008)	Germany	arable	72%	8	3	4	15	15%		

RESPONSE SHEET – REVIEWER #2

Reitmayr (1995)	Germany	wheat, potatoe	63%	6	1	1	8	8%
Tuomisto (2012)	EU	arable	36%	9	2	5	16	16%
			50%				98	100%
			58%					
Animal-based								
Basset-Mens; Werft (2005)	France	pig	95%	8	7	6	21	35%
Casey; Holden (2006)	Ireland	beef	82%	8	3	10	21	35%
Flessa et al. (2002)	Germany	beef/cattle	73%	7	5	6	18	30%
			84%				60	100%
			150%					
Dairy								
Bos et al. (2014)	Netherlands	dairy	61%	9	3	4	10	21%
Dalgaard et al. (2006)	Denmark	dairy	57%	8	2	6	16	21%
Haas et al. (2001)	Germany	dairy	67%	7	8	5	20	32%
Thomassen et al. (2008)	Netherlands	dairy	65%	8	10	6	24	26%
			63%				70	100%
			96%					

Table 2: Determining the emission-difference ($D_{org/conv}$) between organic and conventional production in different countries' contexts through the application of meta-analytical methods; arrows represent the yield/productivity difference for each category, this difference is then multiplied with the emission-difference per ha to derive the emission difference per kg (in bold); PY = publishing year, CY = yearly citations, SJR = SciMago journal ranking, SUM = sum of all three factors, WEIGHT = weighted sums of category. A more detailed explanation of the studies' specifics including the weighting scheme can be found in the SI section (SI.1). Source data are provided as a source data file.

We also created another table with additional information, which we put in the Supplementary Information section (please see SI.1).

Table 3: As with table 1 and 2, we changed the style of the table and added the according symbols and indices to the presented data. We also improved the textual information about the table by linking its content to the method section.

Subsection 3.2, p. 10 f.:

Broad categories [b]	prod. method						Narrow categories [n]	prod. method					
	Conv.			Org.				Conv.			Org.		
	$pp_{b,conv}$ (€/kg Prod)	$C_{b,conv}$ (€/kg Prod)	$\Delta_{b,conv}$ with LUC	$pp_{b,conv}$ (€/kg Prod)	$C_{b,org}$ (€/kg Prod)	$\Delta_{b,org}$		$pp_{b,n,conv}$ (€/kg Prod)	$C_{b,n,conv}$ (€/kg Prod)	$\Delta_{b,n,conv}$ with LUC	$pp_{b,n,org}$ (€/kg Prod)	$C_{b,n,org}$ (€/kg Prod)	$\Delta_{b,n,org}$
Plant-based	0.14	0.04	25%	0.39	0.02	5%	Vegetables	0.69	0.01	1%	1.10	~0.00	~0%
							Fruit	0.50	0.05	9%	0.57	0.03	5%
							Cereal	0.09	0.07	72%	0.31	0.04	12%
							Root Crops	0.08	0.01	14%	0.35	0.01	2%
							Legumes	0.02	0.01	33%	0.13	0.00	3%
							Oilseed	0.37	0.18	50%	0.42	0.11	25%

Animal-based	1.66	1.60 (2,41)	97% (146%)	3.41	2.40	70%	Eggs	1.21	0.22 (0,23)	18% (19%)	3.42	0.33	10%
							Poultry	1.72	2.37 (2,85)	137% (165%)	2.31	3.55	153%
							Ruminants	3.38	4.47 (6,65)	132% (197%)	3.90	6.70	172%
							Pork	1.35	1.00 (1,72)	74% (128%)	3.61	1.49	41%
Milk	0.26	0.21 (0,29)	78% (108%)	0.48	0.20	41%	Milk	0.26	0.21 (0,29)	78% (108%)	0.48	0.20	41%

Table 3: Producer prices (pp), external costs (C) and percentage price increases (Δ) for narrow and broad food categories when externalities resulting from GHG emissions are monetized; producer prices are calculated by dividing the total amount of producer proceeds for each category (in Euro)⁴⁰ with its total production quantity^{36,41}; external costs are derived by multiplying emission values from table 1 with the emission cost rate of 180€/tCO₂eq; percentage price increases are the ratio of external costs to producer prices; in brackets are the values with LUC-emission costs included. Source data are provided as a source data file.

Formula 6, 7, 8 and 9: We deleted these formulas and explained the calculations thereof in the text.

Subsection 3.1, p. 9:

“We aggregate GEMIS emission data ($q_{b,n,i,conv}$) to narrow ($e_{b,n,conv}$) and broad categories ($E_{b,conv}$) by multiplying the respective emission data with the shares from table 3 (cf. formula 2). From this conventional emission values we derive emissions for organic production. For narrow as well as broad categories, the respective conventional emission values are multiplied with the applicable emission-differences ‘ $D_{b,org/conv}$ ’ (cf. table 2).”

Formula 10, 11, 12 and 13: We deleted these formulas and explained the calculations thereof in the text

Subsection 6.3.2, p.

To calculate the costs ‘ C_b ’ of category-specific emissions, we multiply the cost rate ‘P’ for CO₂-equivalents with the category-specific emission data ‘ E_b ’ or ‘ $e_{b,n}$ ’ (depending on whether broad or narrow categories are observed). Further, we determine percentage surcharge costs ‘ Δ_b ’ by setting these costs in relation to the producer price ‘ pp_b ’ of the respective food-category: $\Delta_b = \frac{C_b}{pp_b}$ (the calculation is analogous for narrow categories). These surcharge costs represents the price increase necessary to internalize all externalities from GHG emissions for a specific food-category.

Table 4: We changed the style of the table and added the according symbols and indices to the presented data; we also changed the table description with new passages, again, being underlined.

Subsection 6.2.1, p.

production data

Broad categories [b]	share in broad categories	Narrow categories [n]	share in narrow categories	Food-specific [i]	Total production quantity (in 1,000t) [q _{b,n,i,conv.}]
Plant-based	7%	Vegetables	98%	field-vegetables	3,166
			2%	tomatoes	78
	2%	Fruit	100%	other	63
			fruit	1,183	
	33%	Cereal	5%	other	0
			82%	rye	733
			1%	wheat	13,026
			13%	oat	101
			13%	barley	2,080
			100%	other	0
54%			Root Crops	100%	potatoes
1%	Legumes	100%	other	17,837	
1%	Legumes	100%	beans	148	
3%	Oilseed	100%	other	280	
		rapeseed	1,595		
Animal-based	8%	Eggs	100%	other	61
			eggs	716	
	17%	Poultry	100%	other	0
			broilers	1,509	
	13%	Ruminants	100%	other	0
beef			1,099		
62%	Pork	100%	other	18	
		pork	5,559		
Dairy	100%	Milk	100%	other	0
				milk	31,736

Table 4: Production data [q_{b,n,i,conv.}] for food-specific products and share in broad and narrow categories for 2016 in Germany; production data was obtained from the German Federal Office of Statistics ³⁶ and AMI ^{41,70}. Source data are provided as a source data file.

Table 5 & 6 are now Figure 1 & 2: We changed the style of the figures and added the according symbols to the presented data:

Figure 1: Visualization of the monetary costs [C] for broad categories (animal, milk, plant-based in comparison to conventional, organic production) arising from monetized externalities of GHG emissions.

Figure 2: Visualization of the relative percentage price [Δ] increases for broad categories (animal-based, dairy, plant-based in comparison between conventional and organic production) when externalities of GHG emissions are included into the producers price; for conventional production (animal-based and dairy) the surcharge from LUC emissions is highlighted separately. Source data are provided as a source data file.

Figure 2 is now Figure 4: We added the according symbols to the figure in red. By doing so, calculations can be better conceptualized within our methodology:

Figure 4: Visualization of the quantification process and corresponding input and output data. GEMIS data ³⁸ ($g_{b,n,i,conv}$) and

production data ^{37,42,65} ($q_{b,n,i,conv}$) are combined and emission data for broad ($E_{b,conv}$) and narrow ($e_{b,n,conv}$) categories is derived for conventional production. Organic emission values are calculated by multiplication of conventional emission values ($E_{b,org}$ and $e_{b,n,org}$) with the emission difference ($D_{b,org/conv}$) (cf. subsection 6.2.1). Source data are provided as a source data file.

Furthermore, we have now more extensively linked our findings with previous literature. Please find corresponding text in the following (new passages are underlined):

Section 4., p. 14 ff.

“As the results show, the production of animal-based products – especially of meat – causes the highest emissions. These results are in line with the prevailing scientific literature ^{18–21,42}.

[...]

Secondary animal-based products, such as milk and eggs, however, cause lower emissions than meat. Again, these findings are in line with other sources ^{21,44}.

[...]

The feed of organic dairy cows incorporates a significantly higher proportion of grazing (29.5% compared to 0.5%), which also avoids GHG emissions associated with the production of industrial feed compared to conventional dairy cows ⁴⁷. Moreover, the use of grassland instead of farmland leads to the preservation of CO₂ sinks ⁴⁸. However, the difference between farming practices is lower in both primary, and secondary animal-based products compared to the difference in plant farming. This may be explained with the higher land use ^{49–51}, living age and lower productivity of organically raised animals ⁴⁷ (cf. table 2) counterbalancing or even reversing the described positive aspects of organic animal farming.

[...]

Due to price elasticities of demand for food products (which are consistently regarded as ‘normal goods’ in economic literature), appropriate pricing of food would make products of organic production more competitive compared to their conventional counterparts ⁵³: customers would increasingly opt for organic foodstuff due to the lowered price-gap between the two options. This could potentially press the boundaries of land use for agriculture as organic practices mostly require more land than conventional systems due to lower yields ^{54–56}. However, our results suggest an increase in the prices of animal-based products to a significantly larger extent than the prices of plant-based products. The presumed consequential decline of animal-based product consumption would free an enormous landmass currently used for feed-production. Further expansion of area-intensive organic agriculture would subsequently be made possible ⁵⁷. Furthermore, there is evidence that a shift

from conventional to organic practices would indeed be beneficial for the ecosystem services and long-term efficiency provided by the particular land area^{7,58}. If one takes into account the temporal change in yield difference which would result by converting farms from conventional to organic farming, there is scientific consensus that the yield gap will decrease over time^{59,60}. Comparative studies between different cultivation methods also show that organic farming has lower soil-borne GHG emissions and higher rates of carbon sequestration in the soil^{46,61}. Soil degradation resulting from conventional systems would slow down or could even be reversed by changing to organic farming^{62,63}. [...]

Editor

1. To improve the quality of methods and statistics reporting in our papers, we are now asking all authors to **complete an editorial policy checklist that verifies compliance with all required editorial policies**. Please ensure that the checklist is completed and uploaded with your revised article. All points on the policy checklist must be addressed; if needed, please revise your manuscript in response to these points. Please note that this form is a dynamic ‘smart pdf’ and must therefore be downloaded and completed in Adobe Reader, instead of opening it in a web browser.

Editorial policy checklist:

<https://www.nature.com/documents/nr-editorial-policy-checklist.pdf>

Reporting summary:

Reporting and materials availability requirements for Earth sciences research:

<http://www.nature.com/authors/policies/availability.html#requirements>

Response: We have filled out both the policy checklist and the reporting summary and included them in the attachments of this email.

2. In an effort to ensure reproducibility of research data, we now also require that you **provide a separate source data file**. The source data file should, as a minimum, contain the raw data underlying all reported averages in graphs and charts, and uncropped versions of any gels or blots presented in the figures. To learn more about our motivation behind this policy, please see <https://www.nature.com/articles/s41467-018-06012-8>.

Within the source data file, each figure or table (in the main manuscript and in the Supplementary Information) containing relevant data should be represented by a single sheet in an Excel document, or a single .txt file or other file type in a zipped folder. Blot and gel images should be pasted in and labelled with the relevant panel and identifying information such as the antibody used. We also encourage you to include any other types of raw data that may be appropriate. An example source data file is available demonstrating the correct format:

<https://www.nature.com/documents/ncomms-example-source-data.xlsx>

The file should be labelled ‘Source Data’, with the title and a brief description included in your cover letter, and should be mentioned in all relevant figure legends using the template text below:

“Source data are provided as a Source Data file.”

Response: The source data can be found under the following link, which is also referenced in the data availability statement:

https://osf.io/e7v8x/?view_only=0bff6aa858a340df9046816c1404a51c

3. We would like to clarify if and how **the software/algorithms necessary to reproduce the results** will be made available to the scientific community upon publication as required by our material sharing requirements. For more information on this please see <http://www.nature.com/authors/policies/availability.html#code>

In order for the reviewers to evaluate the work adequately they must be able to test the software/review the code themselves. If you have not yet provided the software, we therefore request that you provide a single compressed zip file containing the software with a readme.txt file or other user manual containing complete instructions for installing and running the software. If appropriate, please also provide example data and expected output. Sufficient material should be provided for referees to directly test the performance of the software/algorithm.

If the software and materials are small enough to fit in a single compressed zip file less than 6MB in size, you may email this file directly to me. If the zip file is between 6 MB and 200 MB you may upload it to our file transfer site. If necessary, a second zip file up to 200 MB in size can be used to supply the example data. Please let me know if you need to use this option and I'll send you further details.

Please also fill out and return to me the code and software submission checklist that will be made available to editors and reviewers during manuscript assessment. Please note that this form is a dynamic 'smart pdf' and must therefore be downloaded and completed in Adobe Reader, instead of opening it in a web browser.

<https://www.nature.com/documents/nr-software-policy.pdf>

Response: We did not use any custom code for our study. According to the editorial policy checklist, the 'code availability statement' does not apply to our study. Emission values were derived from the publicly available material flow analysis tool GEMIS (Version 4.95), which can be downloaded here: <http://iinas.org/qemis-download-121.html>. User manuals, system requirements, demos, installation guides and instructions are all provided on this website.

4. **Data availability statements and data citations policy:** All Nature Communications manuscripts must include a section titled "Data Availability" as a separate section after the Methods section but before the References. For more information on this policy, and a list of examples, please see

<https://www.nature.com/documents/nr-data-availability-statements-data-citations.pdf>

- Accession codes for deposited data
- Other unique identifiers (such as DOIs and hyperlinks for any other datasets)
- At a minimum, a statement confirming that all relevant data are available from the authors

- If applicable, a statement regarding data available with restrictions
- If a dataset has a Digital Object Identifier (DOI) as its unique identifier, we strongly encourage including this in the Reference list and citing the dataset in the Data Availability Statement.
- If a source data file is provided, please add a reference to this in the data availability statement. For example:
 - “The source data underlying Figs 1a, 2a–d, 6d, h and 7c and Supplementary Figs 1a and 5d are provided as a Source Data file.”

Response: We added the following data availability statement:

“The datasets generated and analyzed during the current study are available in the Center for Open Science repository, https://osf.io/e7v8x/?view_only=0bff6aa858a340df9046816c1404a51c. The source data underlying Table 1-4, Figure 1,2 and Supplementary Table 5 are provided in this source data file.

Emission values were derived from the publicly available material flow analysis tool GEMIS (Version 4.95), which can be downloaded here: <http://iinas.org/gemis-download-121.html>”

5. DATA SOURCES: We strongly encourage authors to **deposit all new data and code associated with the paper in a persistent repository** where they can be freely and enduringly accessed. Please note that for some data types, deposition in a public repository is mandatory (<http://www.nature.com/sdata/policies>). We recommend submitting the data to discipline-specific, community-recognized repositories, where possible and a list of recommended repositories is provided here: <http://www.nature.com/sdata/policies/repositories>
- If a community resource is unavailable, data can be submitted to generalist repositories such as figshare (<https://figshare.com/>) or Dryad Digital Repository (<http://datadryad.org/>). Please provide a unique identifier for the data (for example a DOI or a permanent URL) in the data availability statement, if possible. If the repository does not provide identifiers, we encourage authors to supply the search terms that will return the data. For data that have been obtained from publicly available sources, please provide a URL and the specific data product name in the data availability statement. Data with a DOI should be included in the reference list and cited where relevant.
- Alternatively, include the data and code in the Supplementary Information. For datasets for which mandatory deposition is not required and the data or code can only be shared on request, please explain why in your Data Availability Statement and in your cover letter.
- Please refer to our data policies here: <http://www.nature.com/authors/policies/availability.html>
- If you opted into the journal hosting details of a preprint version of your manuscript

via a link on our dedicated website (<https://nature-research-under-consideration.nature.com>), it will remain on this site while you are revising your manuscript, as we consider the file to remain active. Should you wish to remove these details, please email naturecommunications@nature.com indicating your manuscript number and the link on our website that was previously sent to you. Please see our pre-publicity policy at <http://www.nature.com/authors/policies/confidentiality.html> For more information, please refer to our FAQ page at <https://nature-research-under-consideration.nature.com/posts/19641-frequently-asked-questions>

Response: All relevant data of the study can be accessed here:
https://osf.io/e7v8x/?view_only=0bff6aa858a340df9046816c1404a51c

6. Springer Nature encourages **all authors and reviewers to adopt an Open Researcher and Contributor Identifier (ORCID)**. ORCID is a community-based initiative that provides an open, non-proprietary and transparent registry of unique identifiers to help disambiguate research contributions. All authors who link their ORCID to their account in our submission system will have their ORCID published on their articles, if the article is accepted for publication. Please note that this is only possible if ORCIDs are linked prior to acceptance, that is, it is not possible to add ORCIDs at proof. Please ensure that all co-authors are aware that they can add their ORCIDs to their accounts and that they must do so prior to acceptance.
- To add an ORCID please follow these instructions:
1. From the home page of the MTS click on ‘Modify my Springer Nature account’ under ‘General tasks’.
 2. In the ‘Personal profile’ tab, click on ‘ORCID Create/link an Open Researcher Contributor ID (ORCID)’. This will re-direct you to the ORCID website.
 - 3a. If you already have an ORCID account, enter your ORCID email and password and click on ‘Authorize’ to link your ORCID with your account on the MTS.
 - 3b. If you don’t yet have an ORCID account, you can easily create one by providing the required information and then clicking on ‘Authorize’. This will link your newly created ORCID with your account on the MTS.
- If you experience problems in linking your ORCID, please contact Platform Support

Response: We created ORCID accounts for all authors of this article and linked them to our accounts in the Nature-Submission-System. The links to our ORCID accounts are the following:

Maximilian Pieper: <https://orcid.org/0000-0002-5328-2318>

Amelie Michalke: <https://orcid.org/0000-0002-7380-0231>

Tobias Gaugler: <https://orcid.org/0000-0002-0992-4141>

REVIEWER COMMENTS

Reviewer #1 (Remarks to the Author):

Dear Pieper et al,

I found significant improvement in the clarity of the information presented in the paper and the approach used. My general and specific comments are given below:

General comments:

I find the language to be more clear yet there is scope to make it more clear and even shorter. I have indicated some specific sentences in the MS (attached)

Specific comments:

It is mentioned that for organic farming, EU regulations only allow the animal feed to be grown organically and in the same region. The regional context need to be clear, i.e. is it the same country or could be the entire EU, or there are other specific criteria? Also, it is mentioned later that farm associations such as Bioland put restrictions on feed import from Latin America. It is conflicting because if there is a regional restriction on feed imports, what is the relevance of banning imports from Latin America by individual organisations?

Furthermore, if the feed is allowed to be imported and the restrictions are only for Latin America, does it mean that imports from other parts of the world such as Africa and Asia are allowed or take place? If so, it is important to ascertain that such imports do not have LUC related emissions in those regions before taking LUC out of calculations.

In table 2, What is 'arable'? Looking in the Tuomisto (2012), I guess authors mean there is no specific crop differentiation-is it correct? This needs to be made clear. I also suggest using the citation as reference numbers as it will be more convenient to find relevant references.

Authors give strong rationale that organic farming cause negligible LUC, but at one point they say, 'Moreover, the use of grassland instead of farmland leads to the preservation of CO2 sinks 48. However, the difference between farming practices is lower in both primary, and secondary animal-based products compared to the difference in plant farming. This may be explained with the higher land use 49–51, living age and lower productivity of organically raised animals 47 (cf. table 2) counterbalancing or even reversing the described positive aspects of organic animal farming.' This is conflicting as it indicates that organic animal production has higher LUC? Please make it more clear.

In the following statement, 'In case GEMIS offered no data for Germany for certain foodstuff (this is the case for maize, milk and eggs), data from climatically comparable European countries is used.', you should provide which countries were actually used for different categories. I see that you have listed studies in Tables 2 and in the supplementary material. Are these the countries ultimately used? In such a case, I would find it difficult to compare the climatic conditions of France and Spain with Germany. Hence it is important that you mention the countries and for what food categories those countries were used giving clear rationale.

I would like to see more about working principals behind GEMIS. Since GEMIS covers the emissions related to resource extraction to its processing and transportation, these mechanisms also need to be similar to Germany and not just the climate. Do authors assume that those processes are similar across the EU?

On page 23, I am not able to relate Table 3 with the text (see the marked section in the attached MS). Is it table 4?

On the page 23, 'These shares are expressed in formula 2a and 2b (sub-section 6.3.1) by the terms

$p_{(b,n,conv)}/P_{(b,conv)}$ (share in broad categories) and $q_{(b,n,i,conv)}/p_{(b,n,conv)}$ (share in narrow categories)', I am unable to understand whether it is only for conventional or authors assume that share of each category remains the same for both conventional and organic?

About the selection of the literature for meta-analysis, 1968 to 2018 is a very broad time range given that authors address only one year (2016) in this exercise. Certainly, the older papers such as those published more than 20 years ago would have had significantly different farming conditions and practices as well as the technological processes. Hence, there is a point in assigning greater weightage to more recent studies. Still, I wonder why authors did not select a more narrow time range for selection of literature? It is also important to pay attention to the years during which the data was collected in the respective article Rather than the publication year.

I am not trained enough in economics, hence not well equipped for commenting on cost analysis. I hope that the other reviewer can review it.

Best wishes

Reviewer #2 (Remarks to the Author):

This revised version is much better. The Authors made a great effort.

REVIEWER COMMENTS

Reviewer #1 (Remarks to the Author):

Dear Pieper et al,

I found significant improvement in the clarity of the information presented in the paper and the approach used.

Response: We are very grateful for this feedback and are glad that our extensive efforts during the first revision phase came to fruition. Your detailed and specific feedback, as well as the feedback of Reviewer #2, were very valuable to elevate the quality of our paper. We thank you for your specific inquiry.

My general and specific comments are given below:

General comments:

I find the language to be more clear yet there is scope to make it more clear and even shorter. I have indicated some specific sentences in the MS (attached)

Response: Thank you for acknowledging the progress we have made with the first revision process, also due to your comments and suggestions.

We do agree that the language we use can be even shorter and clearer. According to the information we have received regarding the editing process there will be an inspection of linguistic correctness by the journal later on in the submission process. We have, however, improved (i.e. clarified and shortened) several sentences, as you have indicated. Thank you for pointing those out. You can find many of the regarding passages below, with changes underlined (please find all edits indicated in the manuscript as well):

Abstract, p. 1:

“[...] Available research assessing agricultural external costs lacks a differentiation between farming systems and food categories. [...]”

Section 1., p. 2:

“In their 2019 special-report on ‘Climate Change and Land’ the Intergovernmental Panel on Climate Change (IPCC) calculated a share of 23% of global anthropogenic greenhouse gas (GHG) emissions originating from agriculture forestry and other land use activities.”

Section 1., p. 2:

“In this paper we show a possible application of this economic instrument by calculating the surcharges for foodstuff needed for a proper internalization of external costs from GHG emissions.”

Section 2., p. 4:

“Congruent to methodological differences for monetizing agricultural greenhouse gases, there are also differences in the estimation level of greenhouse gas costs.” [shortened sentence]

Section 2., p.5:

“N₂O is produced in agriculture mainly due to direct emissions from agricultural soils, mostly caused by the overapplication of nitrogen fertilizer, and indirect emissions from the production of such fertilizer³⁴.” [shortened sentence]

Subsection 3.1, p. 9:

“As follows from table 2, with LUC emissions included, organic produced food causes fewer emissions in the broad plant-based and dairy categories, while causing slightly higher emissions in the animal category. In the narrow categories organic production performs worse for eggs, poultry and ruminants.”

Section 4., p. 14:

“This may be explained with the higher use of land due to organic regulations prescribing a certain amount of land per animal, which is higher compared to average conventional production⁴⁶⁻⁴⁸, as well as a higher living age and lower productivity of organically produced feed and raised animals⁶⁰ (cf. table 2)

Subsection 6.2.1, p. 22:

“Thus, we have to focus solely on imported feed for conventional animal-based and dairy products.” [shortened sentence]

Specific comments:

- 1) It is mentioned that for organic farming, EU regulations only allow the animal feed to be grown organically and in the same region. The regional context need to be clear, i.e. is it the same country or could be the entire EU, or there are other specific criteria?

*Response: Thank you for pointing this out. Indeed, the word ‘regional’ can be understood in different ways. We are, however, referring to the definition of the inspection authorities of the German federal states, who have agreed to regard regional farms as those from the same or a directly neighboring federal state or political entity (cf. BÖLW. „Further development of organic legislation on the basis of the existing EU Organic Regulation 834/2007 and its implementing regulations 889/2008 and 1235/2008 Proposed amendment to the law, 2017. [**Subsection 3.1, p. 7**](https://www.topagrar.com/dl/2/7/5/7/0/5/9/170607_BOeLW_Vorschlaege>Weiterentwicklung Bio-Recht.pdf.) It should be noted, that Germany consists of 16 federal states. In this context, region can thus be understood as a fairly small geographical area. As you pointed out correctly, this was not made clear within the manuscript so far. We have now clarified this, with respective sections now reading as follows (changes are underlined):Name	name (in GEMIS)	category (in GEMIS)	production-level (in GEMIS)	Code	CO2- eq
field-vegetables	Feldgemüse	Gemüse	Anbau	AnbauFeldgemüse-DE-2010	0.0328
Tomatoes	Tomate	Gemüse	Anbau	AnbauTomate-DE-2010	0.3943
Fruit	Obst	Obst	Anbau	AnbauObst-DE-2010	0.2531

Rye	Roggen	Getreideerzeugnisse	Anbau	AnbauRoggen-DE-2010	0.2204
Wheat	Weizen-Körner	Getreideerzeugnisse	Anbau	AnbauWeizen-Körner-DE-2010	0.3757
Oat	Hafer	Getreideerzeugung	Anbau	AnbauHafer-DE-2010	0.3605
Barley	Gerste	Getreideerzeugung	Anbau	AnbauGerste-DE-2010	0.3335
Potatoes	Kartoffeln	Gemüse	Anbau	AnbauKartoffel-DE-2010	0.0648
Beans	Bohnen	Gemüse	Anbau	AnbauFeldgemüse-DE-2010	0.0328
Rapeseed	Rapsöl	Raps-Öl	NG-Herstellung	NG-HerstellungRapsöl-DE-2010	1.0192
Eggs	Eier	Eier	Tierhaltung Legehennen	TierhaltungLegehennen(Ei)-DE-2010	1.1711
Broilers	Masthähnchen	Fleisch	NG-Schlachtereier	NG-SchlachtereierDE-Masthähnchen-2010	13.1718
Beef	Rind	Fleisch	NG-Schlachtereier	NG-SchlachtereierDE-Rind-2010	24.8637
Pork	Schwein	Fleisch	NG-Schlachtereier	NG-SchlachtereierDE-Schwein-2010	5.5486
Milk	Milch	Milchprodukte	Tierhaltung Milchkuh	TierhaltungMilchkuh(Milch)-DE-2010	1.0958

8) On page 23, I am not able to relate Table 3 with the text (see the marked section in the attached MS). Is it table 4?

Response: We are sorry for this inattention and thank you for the thorough review. In fact, it should read: "Table 4". We have now corrected the error in the main text.

9) On the page 23, 'These shares are expressed in formula 2a and 2b (sub-section 6.3.1) by the terms $p_{(b,n,conv)}/P_{(b,conv)}$ (share in broad categories) and $q_{(b,n,i,conv)}/p_{(b,n,conv)}$ (share in narrow categories)', I am unable to understand whether it is only for conventional or authors assume that share of each category remains the same for both conventional and organic?

Response: Thank you for this comment. You have a valuable point here, as we probably didn't explain this sufficiently in the text.

Yes, we assume that the share of each category remains the same for both conventional and organic. This is because doing otherwise would not enable a fair comparison between production systems on the aggregation-level of broad categories. For example, beef makes up over 50% of all produced food in the organic animal-based product category, while it only accounts for 25% of the conventional animal-based product category. As beef production produces the highest emissions of all foodstuff, these high emissions would be weighted far stronger in the organic category than in the conventional category. This would create ratios between emission values of organic and conventional broad categories that would not be representative of the ratios between organic and conventional narrow categories. We have already tried to elaborate on this in subsection 6.3.1:

Subsection 6.3.1, p. 29f.:

“Concerning the reasoning behind the method, the question that might come to mind is why the differentiation between farming systems happens after the aggregation and not before. This is due to the fact that the proportional production quantities of specific food as well as food categories to each other differ from conventional to organic production. Let us imagine aggregation would take place after the differentiation of farming systems: For example, beef actually makes up over 50% of all produced food in the organic animal-based product category, while it only accounts for 25% of the conventional animal-based product category (cf. production values in table 3). As beef production produces the highest emissions of all foodstuffs, these high emissions would be weighted far stronger in the organic category than in the conventional category and thereby producing a higher mean for the organic animal-based product category than for the conventional one. As can be seen from this example, the organic animal-based product category could have a higher mean of emissions than the conventional animal-based product category while still having lower emissions for each individual organic animal-based product than conventional production. Deriving GHG emissions of foodstuff before aggregating to broader categories would thus be problematic and create means not representative for the elements that make up the broader category. To prevent this problem, the chosen method in this paper is thus to first aggregate to the chosen level of granularity (broad or narrow food categories) and then to derive emissions of organic production from conventional production data.”

However, as this explanation is embedded in a slightly different context, we also added further explanation on our calculation logic at the passage in the text that was pointed out by you:

Subsection 6.3.1, p. 23:

“Only production data for conventional production is used. Thereby we imply ratios of production quantities across the food categories for organic production that are equal to those of conventional production. This does not fully reflect the current situation of organic production properties but allows for a fair comparison between emission data of organic and conventional food categories. Doing otherwise would create ratios between emission values

of organic and conventional broad categories that would not be representative of the ratios between organic and conventional narrow categories.”

10) About the selection of the literature for meta-analysis, 1968 to 2018 is a very broad time range given that authors address only one year (2016) in this exercise. Certainly, the older papers such as those published more than 20 years ago would have had significantly different farming conditions and practices as well as the technological processes. Hence, there is a point in assigning greater weightage to more recent studies. Still, I wonder why authors did not select a more narrow time range for selection of literature?

Response: The reviewer’s criticism is understandable and has prompted us to adjust the (too) long period (1968 to 2018). As a result, papers that are older than 20 years now receive 0 points in our scoring. This also gives the oldest paper we have considered (Reitmayr, 1995) a value of 0 in this category. This results in small changes in our computation, which we have included in table 2: The emission difference in the "plant based" category (rounded to whole percent) has changed from 58% to 57%. The other changes were so small that they are not visible due to the rounding in table 2. We have also included all resulting changes in our subsequent calculations.

New weighting scheme:

year	numbers of studies	new weighting
2015	xxx	10
2014	x	10
2013		9
2012	xxx	9
2011	x	8
2010		8
2009		7
2008	xx	7
2007		6
2006	xx	6
2005	x	5
2004		5
2003		4
2002	x	4
2001	x	3
2000		3
1999		2
1998		2
1997		1
1996		1
1995	x	0
1994		0

Table 2 (p. 8):

Name	Country	Produce	D _{org/conv}	relevance		
				PY	CY	SJR

Plant-based									
Aguilera et al. (2015a) ⁴⁰	Spain	citrus, fruits	49%	10	3	10	23	26%	
Aguilera et al. (2015b) ⁴¹	Spain	cereals, legumes, veg.	45%	10	3	10	23	26%	
Cooper et al. (2011) ⁴²	UK	crop rotation (no differentiated values for specific crops given)	42%	8	2	2	12	13%	
Küstermann et al. (2008) ⁴³	Germany	arable (no specific crop differentiation/rotation described)	72%	7	3	4	14	16%	
Reitmayr (1995) ⁴⁴	Germany	wheat, potatoe	63%	0	1	1	2	2%	
Tuomisto et al. (2012) ⁴⁵	EU	arable (no specific crop differentiation/rotation described)	36%	9	2	5	16	18%	
			49%				98	100%	
			57%						x 117%
Animal-based									
Basset-Mens; Werft (2005) ⁴⁶	France	pig	95%	5	7	6	18	35%	
Casey; Holden (2006) ⁴⁷	Ireland	beef	82%	6	3	10	19	37%	
Flessa et al. (2002) ⁴⁸	Germany	beef/cattle	73%	4	5	6	15	29%	
			84%				60	100%	
			150%						x 179%
Dairy									
Bos et al. (2014) ⁴⁹	Netherlands	dairy	61%	10	3	4	17	24%	
Dalgaard et al. (2006) ⁵⁰	Denmark	dairy	57%	6	2	6	14	20%	
Haas et al. (2001) ⁵¹	Germany	dairy	67%	3	8	5	16	23%	
Thomassen et al. (2008) ⁵²	Netherlands	dairy	65%	7	10	6	23	33%	
			63%				70	100%	
			96%						x 153%

11) It is also important to pay attention to the years during which the data was collected in the respective article Rather than the publication year.

Response: Thank you for this note. As we do weight the results of all studies with, inter alia, their publication year it is indeed sensible to consider the years during which the data was collected.

We have now included this information in the table SI.1; in the first column we added the year(s) of data collection as well as their average time lag to the respective publication year. Please find the edited version of SI.1 in the following:

Source	Estimation method	Boundaries	Regional coverage ¹	Observed food category	Emission values [kgCO ₂ -eq/ha]		difference org/conv
					conventional	Organic	
Plant-based							49%
Aguilera et al. (2015a) ⁴⁰ p. 719 data collected: N/A	LCA modelling (empiric data from interviews and other studies)	cradle to farmgate	Spain	cereals ² legumes field vegetables ⁴ vegetables greenhouse	1.024 568 3.448 11.841	361 232 1.418 7.592	45% ³
Aguilera et al. (2015b) ⁴¹ p. 730 data collected: 2012 (3)	LCA modelling (empiric data from interviews and other studies)	cradle to farmgate	Spain	citrus fruits ⁵ fruits wine	6.324 2.597 964	1.897 1.480 641	49%
Cooper et al. (2011) ⁴² p. 189 data collected: 2004-2007 (5,5)	empiric data gathered at site	direct and indirect emissions until farmgate; comparable with cradle to farmgate	Nafferton (Northern England), UK	crop rotation ⁶	2.019	841	42%
Küstermann et al. (2008) ⁴³ p. 48 data collected: 1999-2002 (7,5)	modelling (software REPRO)	direct and indirect inputs until farmgate; comparable with cradle to farmgate	Scheyern (Upper Bavaria), Germany	crop rotation ^{7,8}	376	263	70%
Reitmayr et al. (1995) ⁴⁴ (as quoted in Stolze et al. 2000, p. 55) data collected: N/A			Germany	winter wheat potatoes	1.001 1.153	429 958	63%
Tuomisto et al. (2012) ⁴⁵ SI table S.1 data collected: 2001-2008 (7,5)	LCA modelling (data from previous studies and empirical data)	Indirect and direct inputs until farmgate; comparable with cradle to	UK	winter wheat	1.772	629	36%

¹ The specific regional coverage was not stated in all studies. Locations are stated as precisely as possible.

² We have excluded the in underlying study (Aguilera et al. 2011a) observed food category 'rice' for this assessment as it is an irrelevant product for the assessment of the German agricultural sector.

³ When there was more than one food category assessed in one study, we weighted them equally to not interfere with the weighting system between the studies.

⁴ In GEMIS 'field vegetables' constitutes a collective term describing vegetables that are grown in the open air. This form of cultivation is in contrast to the horticultural cultivation of vegetables which uses greenhouses, foil tunnels or other artificially protected areas.

⁵ We have excluded the in underlying study (Aguilera et al. 2011b) observed food categories 'subtropical fruit trees', 'tree nuts', and 'olives' as they are irrelevant products for the assessment of the German agricultural sector.

⁶ Rotation includes winter wheat, potatoes, beans, cabbage, and spring/winter barley.

⁷ Rotation includes potatoes, winter wheat, sunflower, winter rye, and maize.

⁸ Even if sunflower is irrelevant to the assessment of the German foodstuff it is, however, crucial for the underlying crop rotation and farming processes and was therefore not excludable from assessment.

⁹ Tuomisto et al. (2012) and Flessa et al. (2002) explicitly state that the production of farm buildings is not considered. However, as far as it was comprehensible, all other studies have similarly not included assessment of housing production.

		farmgate					
Animal-based							84%
Basset-Mens, Werf (2005) ⁴⁶ data collected: 1996-2001 (6,5)	LCA modelling (data from other studies)	direct and indirect inputs and effects; comparable with cradle to farmgate	France (Bretagne)	pig	4236	4022	95%
Casey, Holden (2006) ⁴⁷ data collected: N/A	LCA modelling (data from questionnaires and other studies)	cradle to farmgate	Ireland	beef	5346	2302	82%
Flessa et al. (2002) ⁴⁸ data collected: 1994-1998 (6)	modelling (on basis of empirical data and other studies)	direct inputs and limited indirect inputs until farmgate; comparable with cradle to farmgate	Germany, Oberbayern (South)	beef/cattle	4177	3037	73%
Dairy							63%
Bos et al. (2007) ⁴⁹ data collected: N/A	modelling (model DairyWise)	indirect and direct emissions; comparable with cradle to farmgate	Netherlands	dairy		11	61%
Dalgaard et al. (2006) ⁵⁰ data collected: 1999 (7)	LCA modelling (based on empirical data from 2138 private farm accounts)	cradle to farmgate	Denmark	dairy; sandy soil	6.335	5.459	57%
				dairy; sandy loam soil	5.803	1.669	
Haas et al. (2001) ⁵¹ data collected: 1998 (3)	LCA modelling (based on empirical data from 35 farms in the region)	direct and upstream (indirect) processes; comparable with cradle to farmgate	Germany, Allgäu (Southern Bavaria)	dairy	9.400	6.300	67%
Thomassen et al. (2008) ⁵² data collected: 2003 (5)	LCA modelling (based on empirical data from field studies of 10 conventional and 11 organic farms, and data of previous studies)	cradle to farmgate	Netherlands	dairy	20.598	13.405	65%

Unfortunately, some studies do not declare their reference years or the years in which field data or experimental data was obtained. We indicate this with “N/A” in table SI.1.

When studies indicate mixed sources for their data, e.g. empirical data and previous literature, we use the years of field experiments or empirical data as the years of data collection. For example, Tuomisto et al. (2012) use results of previous literature for modelling, but use average field data of 2001-2008 from England and Wales for their yield data. We therefore set the years of data collection as 2001-2008.

¹⁰ Production of fertilizer was considered; other indirect inputs for precursors like pesticides and seeds were not included as they were considered negligible; infrastructure (machines and buildings) was not included. This studies system boundaries are least in line with our assessment scope but are still comparable due to the explanation as to why certain processes were excluded.

¹¹ As Bos et al. (2007) reports „GHG emissions per ha on the conventional dairy farms are 65% higher than on the organic model farms.” (p.3). We set organic as 100% and conventional as 165%.

¹² Authors refer to cradle to grave approach when introducing to the topic of LCA. They continue although with cradle to farmgate assessments of nitrogen surpluses, for example. The input data does also not include processes after farmgate. Therefore, we find this approach to be comparable with cradle to farmgate.

The time lag between data collection and publication year of all studies reporting years of data collection lies between 3 and 7.5 years. There is no tendency to be noted whether older studies have a longer time lag or vice versa.

We have now revised our calculations as follows:

- *The weighting of the publications years is now adapted to a weighting of the years of data collection.*
- *If a study does not declare this information it is weighted with the lowest weight in this weighting category (=0).*

The calculation show, that none of the results change significantly or even at all. Biggest changes are of less than 1% (organic dairy surcharge) or 0.02€ (organic ruminant external costs) or 0.11kg CO₂eq/kg Product (organic beef emission). These changes, in our opinion, do not alter the conclusions of our study or significantly improve the quality of our results. We therefore suggest to keep the information about the years of data collection in table SI.1, enhanced according to your suggestion, but do not change the weighting process with these new years as there is no information about this available for some studies (see above).

I am not trained enough in economics, hence not well equipped for commenting on cost analysis. I hope that the other reviewer can review it.

Best wishes

Reviewer #2 (Remarks to the Author):

This revised version is much better. The Authors made a great effort.

Response: *Thanks a lot for acknowledging the effort and significant improvements we have achieved during the first revision phase, also due to your comments and suggestions.*

REVIEWERS' COMMENTS

Reviewer #2 (Remarks to the Author):

This revised version is much better. The Authors made a great effort.

REVIEWER COMMENTS

Reviewer #2 (Remarks to the Author):

This revised version is much better. The Authors made a great effort.

Response: Thank you very much for acknowledging the effort we put into incorporating the greatly valuable remarks from both reviewers. We highly appreciate the detailed comments and suggestions during both revision phases, which enabled us to elevate the quality of our paper to a great extent.